# TIME: Tensor-Factorized Mixture-of-Experts with Intrinsic Routing for Lifelong Multimodal Knowledge Editing

Dexuan Xu[1]  Jieyi Wang[2]  Shijie Li[3]  Hanpin Wang[1]  Yongzhi Cao[1]  Yu Huang[4]

## Abstract

Lifelong multimodal knowledge editing allows vision language models to continuously adapt to dynamic updates to avoid catastrophic forgetting. To mitigate interference between sequential updates, recent paradigms have shifted towards modular parameter isolation. However, this strategy faces a critical scalability bottleneck: accumulating dense parameter blocks can lead to excessive memory growth, and managing these independent modules often uses decoupled routing mechanisms, resulting in architectural redundancy. To address this issue, we propose **TIME** (**T**ensor-Factorized **I**ntrinsic **M**ixture-of-**E**xperts), a unified framework harmonizing parameter efficiency with structural self-routing. TIME parameterizes each knowledge edit as a compact CP-decomposed tensor, significantly reducing complexity compared to low-rank matrices. Furthermore, departing from auxiliary semantic retrievers, we introduce an intrinsic routing mechanism that utilizes the tensor's input factors to directly define the active subspace, effectively enabling expert parameters to serve simultaneously as the routing logic. Extensive experiments demonstrate that TIME achieves state-of-the-art performance on lifelong editing benchmarks while successfully reducing memory usage and inference latency.

## 1. Introduction

Large vision-language models (VLMs) are now the foundation of multimodal artificial intelligence, yet their knowl-

[1]School of Computer Science, Peking University, Beijing, China [2]School of Software and Microelectronics, Peking University, Beijing, China [3]Southern Power Grid Digital Platform Technology Company Limited, Guangzhou, China [4]National Engineering Research Center for Software Engineering, Peking University, Beijing, China. Correspondence to: Yu Huang <hy@pku.edu.cn>.

*Proceedings of the 43rd International Conference on Machine Learning*, Seoul, South Korea. PMLR 306, 2026. Copyright 2026 by the author(s).

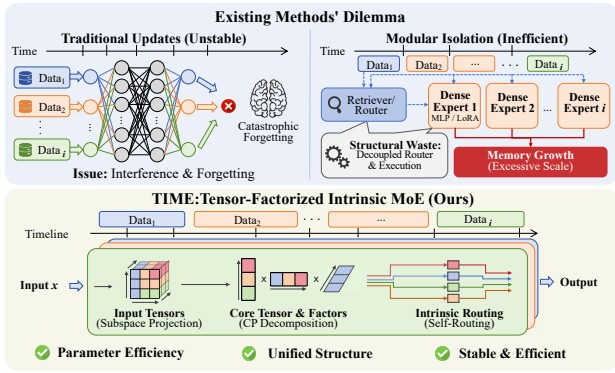

*Figure 1.* **Paradigm Comparison.** Existing editing methods suffer from architectural redundancy and memory bloat due to decoupled routing. TIME introduces a unified framework where CP-decomposed experts inherently determine relevance via subspace projection. This intrinsic routing mechanism eliminates external retrievers, harmonizing parameter efficiency with lifelong stability.

edge remains fixed after training. This makes them prone to errors or hallucinations when the world changes. Therefore, lifelong knowledge editing, defined as the ability to continuously update a model without retraining or forgetting old knowledge, has become a key research topic.

Early methods tried to modify the shared weights directly (e.g., MEND (Mitchell et al., 2022a), ROME (Meng et al., 2022)). However, they often struggle in lifelong settings: as updates accumulate, new edits interfere with old ones, causing the model to forget previous knowledge (Seong et al., 2025). To solve this problem, the main trend has shifted towards modular parameter isolation, such as retrieval-augmented generation (RAG) (Zhang et al., 2025a) or mixture-of-experts (MoE) editors (Chen et al., 2025). By storing each edit in a separate, independent module, these methods effectively avoid interference.

However, this isolation strategy creates a scalability bottleneck. As shown in Figure 1, current methods usually store edits as dense matrices (MLP or LoRA (Hu et al., 2022)), leading to linear memory growth that becomes excessive when scaling to thousands of updates. More importantly, managing this growing library requires a complex and decoupled design: existing methods need an extra retriever to find the right expert. This results in significant structural

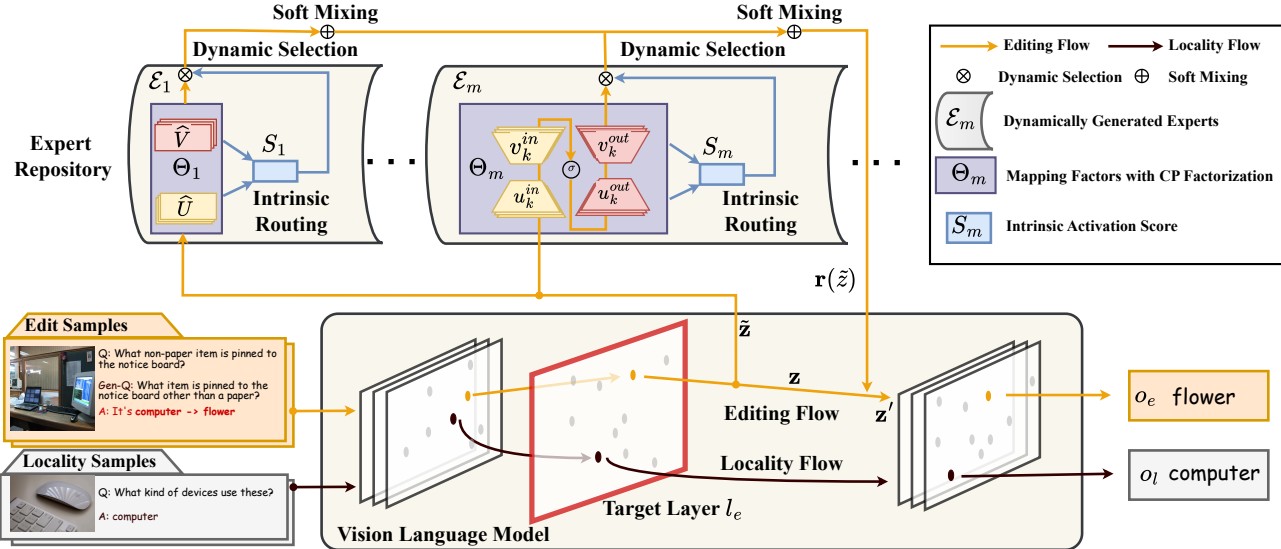

*Figure 2.* Illustration of TIME. Each edit instance is factorized into CP-decomposed low-rank components and stored as experts in a repository. The intrinsic routing mechanism retrieves and fuses relevant experts to enable parameter-efficient editing.

waste, as the system must maintain separate parameters for routing (finding the edit) and execution (applying the edit). The community thus faces a dilemma: traditional updates are unstable, while modular isolation is inefficient.

In this paper, we argue that separating routing from execution is unnecessary. We believe that the parameters defining *how* to edit a representation naturally contain the information about *where* and *when* to apply it. Based on this idea, we propose **TIME** (**T**ensor-Factorized **I**ntrinsic **M**ixture-of-**E**xperts), a unified framework that combines parameter efficiency with self-routing.

Specifically, TIME treats the weight update as a tensor operation and approximates it via CANDECOMP/PARAFAC (CP) decomposition (Kolda & Bader, 2009). This makes the parameters highly compact, reducing memory cost from quadratic to linear. Crucially, we introduce **intrinsic routing**. We observe that the input-side factors of the CP decomposition naturally capture the expert's sensitivity to specific inputs. By treating these factors as an active subspace, we can select the right expert by simply projecting the input onto them. This allows the expert parameters to act as the router themselves, removing the need for external retrievers.

Our contributions are summarized as follows:

- We identify the scalability bottleneck in modular editing and propose TIME, a tensor-based framework that resolves the trade-off between stability and efficiency.

- We introduce intrinsic routing, a parameter-free mechanism that makes routing decisions directly from the expert's tensor structure via subspace projection.

- Extensive experiments show that TIME matches state-of-the-art performance while reducing memory usage and inference latency significantly.

## 2. Preliminary

### 2.1. Vision Language Models and Edit Instances

Let $\mathcal{V}$ denote the space of images, $\mathcal{P}$ the space of textual prompts, and $\mathcal{O}$ the space of textual outputs. A pretrained vision language model is a conditional generative model $f_\theta : \mathcal{V} \times \mathcal{P} \to \mathcal{O}$.

A knowledge edit instance is defined as a triplet:

$$e = (v_e, p_e, o_e), \qquad (1)$$

where $v_e \in \mathcal{V}$, $p_e \in \mathcal{P}$, and $o_e \in \mathcal{O}$ represents the target output. We assume the base model prediction is incorrect or outdated, i.e., $\arg\max_o p_\theta(o \mid v_e, p_e) \neq o_e$. The goal of a knowledge editor is to produce an updated model $f_{\theta'}$ that satisfies the target prediction while maintaining *Locality* on unrelated knowledge and *Generality* on relevant inputs (see Appendix B.1 for metric definitions).

### 2.2. Lifelong Editing with Modular Repository

In the lifelong editing setting, the model encounters a continuous stream of edit instances $\mathcal{D} = \{e_t\}_{t=1}^T$ arriving sequentially. The editor must update the model $f_{\theta_{t-1}}$ to $f_{\theta_t}$ without accessing the full historical data. To mitigate catastrophic forgetting, recent approaches adopt a modular editing strategy (Chen et al., 2025). Instead of modifying the global parameters $\theta$ directly, a persistent expert repository is main-

tained:

$$\mathcal{E} = \{\Theta_m\}_{m=1}^{M}, \tag{2}$$

where each expert $\Theta_m$ encapsulates a specific knowledge update. For an incoming query, the system retrieves a subset of relevant experts to intervene in the inference process.

## 3. TIME

We propose **TIME** (**T**ensor-Factorized **I**ntrinsic **M**ixture-of-**E**xperts), a scalable framework that constructs low-rank map of retrieved block edits through CP decomposition. Unlike previous approaches that rely on external retrieval modules, TIME employs an *intrinsic routing* mechanism where the routing decision is induced directly from the expert's tensor structure. During inference, edits are sparsely activated based on subspace projection and injected into selected layers as controllable perturbations. The schematic diagram is shown in Figure 2.

### 3.1. Problem Setting and Expert Repository

We cast knowledge editing as injecting a data-dependent residual mapping at a chosen intermediate layer. Given a target layer index $l_e$ and the hidden state of a token $\mathbf{z} \in \mathbb{R}^H$, we first apply layer normalization and then add a small context-dependent residual to the layer output:

$$\tilde{\mathbf{z}} = \mathrm{LN}(\mathbf{z}), \qquad \mathbf{z}' = \mathbf{z} + \mathbf{r}(\tilde{\mathbf{z}}), \tag{3}$$

where $\mathrm{LN}(\cdot)$ denotes the layer-normalization operator, $\tilde{\mathbf{z}}$ is its output, $\mathbf{r}(\cdot) : \mathbb{R}^H \to \mathbb{R}^H$ is the residual mapping constructed in the next sections, $\mathbf{z}'$ is the edited layer output, and $H$ is the hidden dimension.

An edit instance $e$ is given by a triplet $(v_e, p_e, o_e)$ of image, prompt, and desired target. We run the base model and extract the intermediate sequence representation at layer $l_e$, $\mathbf{X}_e \in \mathbb{R}^{L \times H}$. This representation supplies evidence for constructing mapping factors.

To support lifelong accumulation, we maintain a persistent expert repository $\mathcal{E}$. Distinct from prior work, our experts do not require auxiliary routing keys (e.g., visual/textual features). Each expert $m$ consists solely of Mapping factors: we parameterize a single-expert in-layer mapping via a vector-level CP factorization. We split the hidden size as $H = s_1 s_2$ with $s_1, s_2 \in \mathbb{N}^+$, and store paired short vectors for both input and output sides at rank $R$:

$$\Theta_m = \left\{ (u_k^{\mathrm{in}}, v_k^{\mathrm{in}}), (u_k^{\mathrm{out}}, v_k^{\mathrm{out}}) \right\}_{k=1}^{R}, \tag{4}$$

$$u_k^{(\cdot)} \in \mathbb{R}^{s_1}, \ v_k^{(\cdot)} \in \mathbb{R}^{s_2}. \tag{5}$$

These factors are used for both the rank-1 reconstruction of $\mathbf{r}(\cdot)$ and the intrinsic routing decision. The repository is $\mathcal{E} = \{\Theta_m\}_{m=1}^{M}$, which grows over time.

### 3.2. CP-Decomposition View of the In-Layer Edit

We formalize the in-layer editing operator as a low-rank structured mapping from an input representation to an output representation. Given a hidden vector $\tilde{\mathbf{z}} \in \mathbb{R}^H$, we first reshape it into a tensor:

$$H = s_1 s_2, \qquad \tilde{\mathbf{z}} \longleftrightarrow Z = \mathrm{reshape}(\tilde{\mathbf{z}}) \in \mathbb{R}^{s_1 \times s_2}. \tag{6}$$

Any linear mapping from $Z \in \mathbb{R}^{s_1 \times s_2}$ to $Y \in \mathbb{R}^{s_1 \times s_2}$ can be represented by a fourth-order mapping tensor $\mathcal{T} \in \mathbb{R}^{s_1 \times s_2 \times s_1 \times s_2}$:

$$Y = \mathcal{T} \times_3 Z, \tag{7}$$

where $\times_3$ denotes contraction over the last two modes of $\mathcal{T}$. To reduce complexity, we approximate $\mathcal{T}$ via CP decomposition:

$$\mathcal{T} = \sum_{k=1}^{R} u_k^{\mathrm{out}} \circ v_k^{\mathrm{out}} \circ u_k^{\mathrm{in}} \circ v_k^{\mathrm{in}}, \tag{8}$$

where $\circ$ is the outer product, $u_k^{\mathrm{out}}, u_k^{\mathrm{in}} \in \mathbb{R}^{s_1}$, $v_k^{\mathrm{out}}, v_k^{\mathrm{in}} \in \mathbb{R}^{s_2}$, and $R$ is the CP rank. Each quadruple $(u_k^{\mathrm{out}}, v_k^{\mathrm{out}}, u_k^{\mathrm{in}}, v_k^{\mathrm{in}})$ defines one rank-1 edit component.

**Matricized Form.** Grouping output and input sides, the mapping tensor can be written compactly via the Khatri–Rao product:

$$\mathcal{T}_{(o|i)} = (U^{\mathrm{out}} \odot V^{\mathrm{out}})(U^{\mathrm{in}} \odot V^{\mathrm{in}})^\top, \tag{9}$$

where $U^{(\cdot)} \in \mathbb{R}^{s_1 \times R}$ and $V^{(\cdot)} \in \mathbb{R}^{s_2 \times R}$. This form expresses the full tensor through paired rank-1 factors, achieving efficient parameterization and compactness.

**Learning Rank Factors.** In practice, these factors are generated dynamically from the contextual representation of the edit sample $\mathbf{X}_e$ at layer $l_e$ via shared cross-attention:

$$H_{\mathrm{att}}^{(\cdot)} = \mathrm{CA}\left(\Phi^{(\cdot)}, \mathrm{LN}(\mathbf{X}_e)\right) \in \mathbb{R}^{R \times d_{\mathrm{mid}}}, \tag{10}$$

where $\mathrm{CA}(\cdot, \cdot)$ denotes cross-attention, $\Phi^{(\cdot)} \in \mathbb{R}^{R \times d_\Phi}$ are rank-specific learnable query templates. Each row of $H_{\mathrm{att}}^{(\cdot)}$ is projected into short vectors:

$$u_k^{(\cdot)} = W_u^{(\cdot)} h_k^{(\cdot)} \in \mathbb{R}^{s_1}, \qquad v_k^{(\cdot)} = W_v^{(\cdot)} h_k^{(\cdot)} \in \mathbb{R}^{s_2}. \tag{11}$$

**Rank-1 Contraction and Reconstruction.** Each component performs an input-side scalar contraction and an output-side reconstruction:

$$s_k(Z) = \sigma\left((u_k^{\mathrm{in}})^\top (Z \, v_k^{\mathrm{in}})\right), \qquad B_k^{\mathrm{out}} = u_k^{\mathrm{out}} v_k^{\mathrm{out}\top}, \tag{12}$$

where $\sigma$ is an activation function. The reconstructed output of a single expert $m$ is:

$$Y_m(Z) = \frac{1}{\alpha\sqrt{R}} \sum_{k=1}^{R} s_k(Z) \, B_k^{\mathrm{out}} \in \mathbb{R}^{s_1 \times s_2}. \tag{13}$$

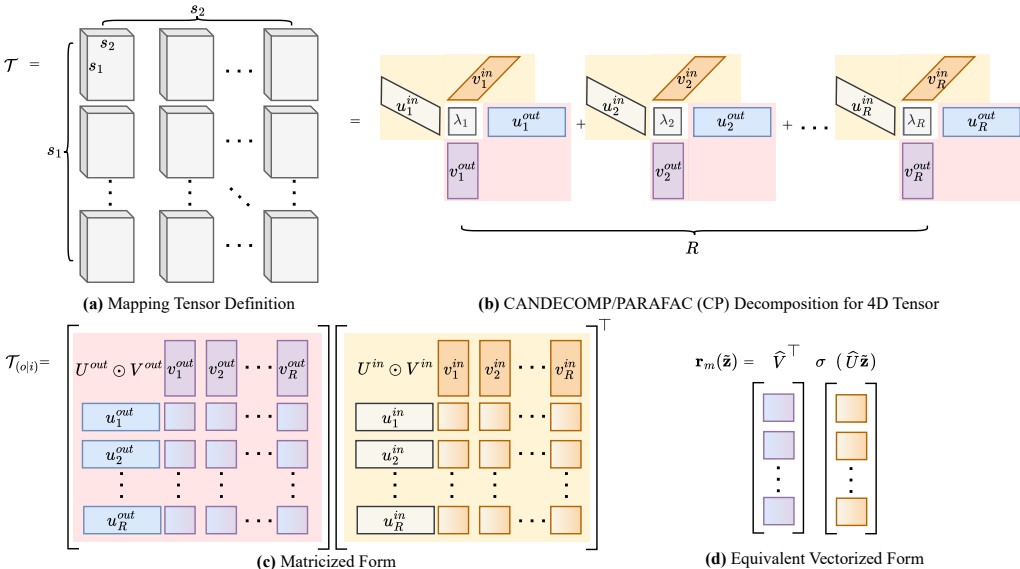

*Figure 3.* Illustration of CP-Decomposition View of the In-Layer Edit.

Vectorizing $Y_m(Z)$ yields the residual:

$$\mathbf{r}_m(\tilde{\mathbf{z}}) = \text{vec}(Y_m(Z)) \in \mathbb{R}^H. \quad (14)$$

**Equivalent Vectorized Form.** For efficiency, each rank-1 basis is flattened into a vector:

$$\mathbf{w}_k^{(\cdot)} = \text{vec}(u_k^{(\cdot)} \otimes v_k^{(\cdot)}) \in \mathbb{R}^H, \quad (15)$$

and stacked with global normalization to form $\widehat{U}$ and $\widehat{V} \in \mathbb{R}^{R \times H}$. Finally, the forward computation of one expert $m$ can be expressed compactly as:

$$\mathbf{r}_m(\tilde{\mathbf{z}}) = \widehat{V}^\top \sigma(\widehat{U}\,\tilde{\mathbf{z}}), \quad (16)$$

indicating that each expert behaves as a two-layer low-rank perceptron with CP-decomposed structure. The proof of the equivalence of the formulas is given in Appendix D.1, and the entire CP decomposition process is shown in Figure 3.

### 3.3. Structure-Aware Intrinsic Routing

Unlike prior approaches that rely on auxiliary semantic encoders to generate external routing keys, we propose a parameter-free routing mechanism inherent to the CP decomposition structure. We observe that the input-side factors of a CP expert naturally define a specific activation subspace. If an input token $\tilde{\mathbf{z}}$ has a significant projection magnitude onto an expert's input factors, it indicates that the expert is structurally sensitive to this input.

**Intrinsic Activation Score.** Recall from Eq. (16) that the forward computation of expert $m$ involves an input projection $\widehat{U}_m\tilde{\mathbf{z}}$. This projection essentially measures the align-

ment between the input representation and the expert's rank-1 input bases. Instead of learning separate routing parameters, we define the *Intrinsic Activation Score* $S_m(\tilde{\mathbf{z}})$ directly as the magnitude of this projection response:

$$S_m(\tilde{\mathbf{z}}) = \sum_{k=1}^{R} \left| \sigma\left( (u_{m,k}^{\text{in}})^\top Z v_{m,k}^{\text{in}} \right) \right|. \quad (17)$$

A high score $S_m$ implies that the input $\tilde{\mathbf{z}}$ falls into the **active subspace** of the edit, necessitating the expert's intervention. This formulation aligns the routing criterion strictly with the expert's functional contribution.

**Dynamic Threshold Selection.** To ensure computational efficiency, we perform sparse routing based on the intrinsic scores. For a repository of size $M$, we employ a threshold-based filtering strategy to dynamically select relevant experts. Specifically, we retain only those experts whose activation scores exceed a pre-defined threshold $\gamma$:

$$\mathcal{M}_{\text{top}} = \{m \in \{1, \dots, M\} \mid S_m(\tilde{\mathbf{z}}) > \gamma\}. \quad (18)$$

**Input-Dependent Soft Mixing.** The final residual update is a weighted combination of the selected experts. We normalize the intrinsic scores over the selected subset to obtain the mixing weights $\pi_m$:

$$\pi_m = \frac{\exp(S_m(\tilde{\mathbf{z}})/\tau)}{\sum_{j \in \mathcal{M}_{\text{top}}} \exp(S_j(\tilde{\mathbf{z}})/\tau)}, \quad \text{for } m \in \mathcal{M}_{\text{top}}, \quad (19)$$

where $\tau$ is a temperature hyperparameter. The final expert injection is then computed as:

$$\mathbf{r}(\tilde{\mathbf{z}}) = \sum_{m \in \mathcal{M}_{\text{top}}} \pi_m \mathbf{r}_m(\tilde{\mathbf{z}}). \quad (20)$$

*Table 1.* A comparison with previous work on E-VQA and E-IC. Results are shown for single editing and lifelong editing (gap=1000).

| Backbone | #Gap | Editor | E-VQA | | | | | | E-IC | | | | | |
|---|---|---|---|---|---|---|---|---|---|---|---|---|---|---|
| | | | Rel. | T-Gen. | I-Gen. | T-Loc. | I-Loc. | Avg. | Rel. | T-Gen. | I-Gen. | T-Loc. | I-Loc. | Avg. |
| BLIP2 | 1 | FT-L | 52.86 | 48.80 | 32.94 | 98.24 | 94.27 | 65.42 | 45.02 | 44.47 | 40.72 | 99.02 | 98.27 | 65.50 |
| | | FT-V | 91.70 | 87.24 | 33.30 | **100.00** | 85.22 | 79.49 | 67.14 | 61.76 | 43.34 | **100.00** | 96.76 | 73.80 |
| | | MEND | 93.13 | 92.76 | 93.07 | 92.00 | 75.81 | 89.35 | **94.96** | **92.45** | **92.33** | 94.95 | 88.86 | **92.71** |
| | | SERAC | 88.39 | 84.50 | 84.25 | 85.82 | 26.00 | 73.79 | 88.71 | 83.81 | 84.38 | 84.28 | 24.70 | 73.18 |
| | | RECIPE | 89.42 | 86.24 | 87.53 | 99.87 | 89.16 | 90.45 | 85.20 | 81.44 | 82.71 | **100.00** | 94.59 | 88.79 |
| | | LEMoE | 93.56 | 92.23 | 91.40 | 98.50 | 85.21 | 92.18 | 93.07 | 91.37 | 83.28 | 94.45 | 60.44 | 84.52 |
| | | LiveEdit | **96.67** | 94.20 | **93.82** | **100.00** | 96.94 | **96.94** | 80.60 | 80.12 | 76.88 | **100.00** | **100.00** | 87.52 |
| | | Ours | 96.55 | **94.65** | 92.34 | **100.00** | **99.98** | 96.70 | 82.21 | 80.44 | 75.82 | **100.00** | **100.00** | 87.69 |
| | 1000 | FT-L | 45.10 | 34.62 | 35.42 | 48.42 | 41.24 | 40.96 | 53.61 | 48.81 | 45.92 | 52.45 | 59.09 | 51.98 |
| | | FT-V | 40.40 | 31.46 | 27.85 | **100.00** | 27.44 | 45.43 | 48.24 | 45.55 | 43.03 | **100.00** | 23.76 | 52.12 |
| | | MEND | 15.84 | 14.35 | 17.73 | 91.74 | 70.17 | 41.97 | 6.54 | 6.51 | 6.50 | 13.52 | 20.38 | 10.69 |
| | | SERAC | 83.35 | 70.80 | 80.32 | 67.66 | 13.13 | 63.05 | 43.12 | 41.69 | 38.72 | 48.08 | 14.88 | 37.30 |
| | | RECIPE | 84.99 | 74.20 | 82.04 | 96.82 | 87.73 | 85.16 | 43.02 | 41.63 | 38.59 | 99.68 | 92.96 | 63.18 |
| | | LEMoE | 19.73 | 17.34 | 18.22 | 72.01 | 31.06 | 31.67 | 43.46 | 43.34 | 37.69 | 93.27 | 67.52 | 57.06 |
| | | LiveEdit | 94.42 | 91.98 | **84.65** | **100.00** | 97.38 | 93.69 | 72.86 | 70.34 | **67.92** | **100.00** | **100.00** | 82.22 |
| | | Ours | **94.92** | **93.59** | 83.29 | **100.00** | 98.93 | **94.15** | **74.24** | **72.58** | 65.81 | **100.00** | **100.00** | **82.53** |
| LLaVA-v1.5 | 1 | FT-L | 93.88 | 87.98 | 80.25 | 99.61 | 94.78 | 91.30 | 73.48 | 72.98 | 65.79 | 99.28 | 99.06 | 82.12 |
| | | FT-V | 87.29 | 76.11 | 53.23 | **100.00** | 96.95 | 82.72 | 56.19 | 56.55 | 49.94 | **100.00** | **100.00** | 72.54 |
| | | MEND | 91.23 | 90.05 | **91.29** | 91.02 | 90.22 | 90.76 | 92.82 | **91.81** | 90.59 | 96.38 | 93.69 | **93.06** |
| | | SERAC | 89.33 | 83.72 | 84.97 | 82.05 | 23.78 | 72.77 | 88.18 | 81.03 | 85.61 | 84.01 | 28.58 | 73.48 |
| | | RECIPE | 91.37 | 86.51 | 87.73 | 94.27 | 88.88 | 89.75 | 84.45 | 76.97 | 81.57 | 96.53 | 96.37 | 87.18 |
| | | LEMoE | 93.60 | 92.77 | 89.99 | 99.28 | 96.98 | 94.52 | **93.80** | 91.42 | **90.61** | 95.14 | 93.00 | 92.79 |
| | | LiveEdit | 94.28 | 94.51 | 88.01 | **100.00** | **100.00** | **95.36** | 82.16 | 81.01 | 78.27 | **100.00** | **100.00** | 88.29 |
| | | Ours | **94.62** | **94.75** | 86.80 | **100.00** | **100.00** | 95.23 | 81.23 | 82.92 | 80.33 | **100.00** | **100.00** | **88.90** |
| | 1000 | FT-L | 71.39 | 59.83 | 57.41 | 55.55 | 48.99 | 58.63 | 59.78 | 54.99 | 54.17 | 65.37 | 78.96 | 62.65 |
| | | FT-V | 69.57 | 56.34 | 44.07 | **100.00** | 41.47 | 62.29 | 49.21 | 47.75 | 43.81 | **100.00** | 35.14 | 55.18 |
| | | MEND | 0.04 | 0.05 | 0.05 | 0.08 | 0.09 | 0.06 | 54.39 | 54.14 | 50.99 | 83.87 | 80.60 | 64.80 |
| | | SERAC | 85.57 | 75.58 | 82.01 | 62.46 | 15.69 | 64.26 | 52.93 | 53.44 | 49.01 | 49.91 | 16.65 | 44.39 |
| | | RECIPE | 87.00 | 76.81 | 83.09 | 86.95 | 87.03 | 84.18 | 53.11 | 53.48 | 48.99 | 87.93 | 94.84 | 67.67 |
| | | LEMoE | 30.80 | 25.75 | 24.32 | 71.45 | 46.23 | 39.71 | 34.50 | 31.38 | 28.14 | 82.09 | 75.88 | 50.40 |
| | | LiveEdit | 92.93 | 90.16 | **84.30** | **100.00** | 96.43 | 92.76 | 72.80 | 69.95 | 57.05 | **100.00** | **99.79** | 79.92 |
| | | Ours | **93.92** | **91.41** | 83.47 | **100.00** | 96.92 | **93.14** | **73.32** | **71.69** | **61.04** | **100.00** | 99.52 | **81.12** |

This intrinsic routing strategy offers two key benefits: (1) **Parameter Efficiency**: It removes the need for storing routing keys and feature extractors. (2) **Consistency**: The routing decision is derived from the exact same parameters used for the edit. This convex combination provides stability and the proof is given in Appendix D.2.

### 3.4. Training Objective via Discriminative Subspace Alignment

Since our routing mechanism relies on the intrinsic projection magnitude of expert parameters rather than external auxiliary networks, we redesign the training objective to jointly optimize the editing performance and the discriminative power of the CP factors. The total loss $\mathcal{L}$ consists of an editing loss and a structure-aware alignment loss:

$$\mathcal{L} = \mathcal{L}_{\text{task}} + \lambda_{\text{align}}\mathcal{L}_{\text{align}}. \tag{21}$$

**Task-Specific Editing Loss.** To ensure the edited model $f_{\theta'}$ incorporates new knowledge while preserving existing capabilities, we retain the standard reliability ($\mathcal{L}_{\text{rel}}$), generality ($\mathcal{L}_{\text{gen}}$), and locality ($\mathcal{L}_{\text{loc}}$) objectives:

$$\mathcal{L}_{\text{task}} = \lambda_{\text{rel}}\mathcal{L}_{\text{rel}} + \lambda_{\text{gen}}\mathcal{L}_{\text{gen}} + \lambda_{\text{loc}}\mathcal{L}_{\text{loc}}. \tag{22}$$

These loss components guide the optimization of the output probability distribution, aligning the model with the target knowledge for the edit instance and its paraphrases while minimizing interference on the locality neighborhood. The details of these losses are shown in Appendix B.2.

**Structure-Aware Alignment Loss.** A critical requirement for Intrinsic Routing is that the target expert $m^*$ (the one currently being added or updated) must exhibit a significantly higher activation score $S_{m^*}(\tilde{\mathbf{z}})$ than other experts in the repository for the relevant input. To enforce this, we introduce an Intrinsic Alignment Loss based on a contrastive formulation.

Given an edit batch with input latent $\tilde{\mathbf{z}}$, let $\mathcal{M}_{\text{neg}}$ be a set of sampled negative experts from the repository. We maximize the normalized activation probability of the target expert $m^*$:

$$\mathcal{L}_{\text{align}} = -\log \frac{\exp(S_{m^*}(\tilde{\mathbf{z}})/\tau)}{\exp(S_{m^*}(\tilde{\mathbf{z}})/\tau) + \sum_{j \in \mathcal{M}_{\text{neg}}} \exp(S_j(\tilde{\mathbf{z}})/\tau)}.$$

Minimizing $\mathcal{L}_{\text{align}}$ explicitly optimizes the subspace orientation of the CP factors. It encourages the input factors $U_{m^*}^{\text{in}}, V_{m^*}^{\text{in}}$ to align their principal directions with the input features, thereby ensuring that the simple projection operation $S_m(\tilde{\mathbf{z}})$ serves as a discriminative and robust proxy for expert relevance.

# 4. Experiments and Analysis

In this section, we conduct extensive experiments to evaluate the proposed TIME framework. We aim to address the following research questions:

- **RQ1:** How does TIME perform compared to state-of-the-art baselines in both single and lifelong multimodal knowledge editing tasks?

- **RQ2:** Does TIME achieve superior parameter efficiency and faster inference speed compared to existing modular editing paradigms?

- **RQ3:** What is the contribution of the proposed Intrinsic Routing mechanism and CP tensor factorization to the overall performance?

- **RQ4:** How do key hyperparameters (e.g., module dimension, CP rank, and target layer) impact the editing effectiveness?

## 4.1. Experimental Setup

**Datasets.** Following (Chen et al., 2025), we conduct experiments on three multimodal knowledge editing benchmarks: (1) **E-VQA** (Cheng et al., 2023), an edited visual question answering dataset testing factual correction within image–question pairs; (2) **E-IC** (Cheng et al., 2023), an edited image captioning dataset with corrected captions; and (3) **VLKEB** (Huang et al., 2024), a large-scale multimodal editing benchmark with counterfactual knowledge.

**Backbones, Baselines and Metrics.** We conduct experiments on single editing and lifelong editing on two vision language models, including BLIP2 (2.7B) (Li et al., 2023) and LLaVA-v1.5 (7B) (Liu et al., 2023). We follow (Chen et al., 2025) to compare the following knowledge editing methods: FT-L (fine-tune the last layer of LLM), FT-V (fine-tune the last layer of vision encoder), MEND (Mitchell et al., 2022a), SERAC (Mitchell et al., 2022b), TP (Huang et al., 2023b), LTE (Jiang et al., 2024b), RECIPE (Chen et al., 2024a), LEMoE (Wang & Li, 2024) and LiveEdit (Chen et al., 2025). We evaluate the Reliability, T-Generality, I-Generality, T-Locality, and I-Locality of the editing results. Specific definitions of these metrics are provided in Appendix B.1.

**Implementation Details.** All experiments are performed on one NVIDIA A800 GPU with 80 GB. For each model, we use its public pre-trained weights. We set the dimension of CP decomposition $s_1 = s_2 = \sqrt{H}$, the default rank size is 4. The scaling parameter $\alpha = 0.1$. The default target layer $l_e = 21$. See Appendix C.1 for more details.

## 4.2. Performance on Single and Lifelong Editing (RQ1)

To answer **RQ1**, we compare TIME with baseline methods across different editing scenarios. Table 1 presents the detailed results on E-VQA and E-IC benchmarks.

**Superior Lifelong Stability.** The core challenge of lifelong editing lies in mitigating catastrophic forgetting over extended sequences. As shown in Table 1 (Gap=1000), traditional hypernetwork-based methods like MEND suffer from severe performance collapse (e.g., dropping to an average score of 0.06 on LLaVA-v1.5 E-VQA), indicating their inability to isolate sequential updates. Similarly, existing modular methods like TP and SERAC struggle to maintain long-term stability due to interference between retrieved parameters.

In contrast, TIME demonstrates remarkable robustness. On the E-VQA benchmark using BLIP2, TIME achieves the highest average score of 94.15, surpassing the strongest baseline LiveEdit. Notably, on the E-IC dataset, TIME achieves an average score of 82.53, outperforming LiveEdit's 82.22. This superior stability is attributed to our intrinsic routing mechanism: by projecting inputs onto distinct tensor subspaces, TIME ensures that new edits are activated only by relevant queries, thereby maximizing the orthogonality between experts and minimizing interference with historical knowledge. As shown in Figure 4, while baselines like FT-V and LEMoE suffer from drastic performance decay as the number of edits increases, TIME maintains consistently high stability across all metrics. This confirms our framework's superior capability in mitigating catastrophic forgetting during long-term continuous editing.

**Competitive Single Editing.** In the single edit setting (Gap=1), capturing new knowledge without disrupting unrelated facts is paramount. TIME remains highly competitive with state-of-the-art methods. On E-VQA (BLIP2), TIME achieves perfect Text-Locality (100.00) and near-perfect Image-Locality (99.98), outperforming fine-tuning methods by a large margin. Although our Reliability scores are marginally lower than LiveEdit, the gap is negligible ($\approx 0.1\%$). This slight trade-off suggests that our CP-decomposed experts, while using fewer parameters, retain nearly the full expressivity of dense matrices. More importantly, the robust locality scores confirm that our intrinsic subspace detection is highly discriminative, preventing the over-editing phenomenon common in other baselines.

**Backbone Generalization across Scales.** We further validate the scalability of TIME across different model architectures and sizes. The performance trends observed on BLIP2 are consistently replicated on the larger LLaVA-v1.5. For instance, in the challenging LLaVA lifelong setting (Gap=1000), TIME maintains an average score of 93.14 on E-VQA and 81.12 on E-IC, effectively outperforming

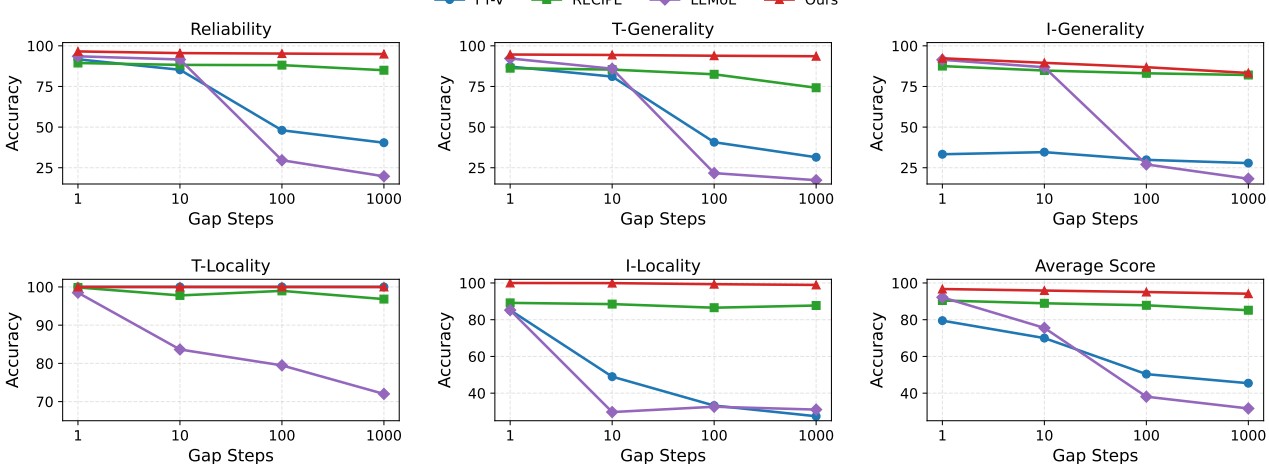

*Figure 4.* Comparison with previous methods when lifelong editing gaps are 1, 10, 100, and 1000.

*Table 2.* Comparison of parameter sizes and editing time on E-VQA between LiveEdit and TIME. #N represents the lifelong editing gap and $H = 1024, r = 4$. Note that TIME achieves lower latency by eliminating external routing modules.

| Backbone | Editor | Params. | Edit Time (s) ↓ | | Avg. Score ↑ | |
|---|---|---|---|---|---|---|
| | | | #1 | #1000 | #1 | #1000 |
| BLIP2 | LiveEdit | 668 MB | 239.25 | 283.73 | **96.94** | 93.69 |
| | **TIME** | **523 MB** | **163.43** | **178.92** | 96.70 | **94.15** |
| LLaVA | LiveEdit | 1178 MB | 248.67 | 776.55 | **95.36** | 92.76 |
| | **TIME** | **786 MB** | **192.28** | **649.83** | 95.23 | **93.14** |

*Table 3.* Ablation studies on E-VQA lifelong editing for BLIP2. To validate our intrinsic design, we ablate: (1) **Selection** (using all experts); (2) **Mixing** (using average instead of score-based weighting); (3) $\mathcal{L}_{align}$ (removing the structure-aware alignment objective); and (4) **Tensor** (using full matrix instead of CP decomposition).

| Methods | Rel. | T-Gen. | I-Gen. | T-Loc. | I-Loc. | Avg. |
|---|---|---|---|---|---|---|
| Fine-tuning | 45.10 | 34.62 | 35.42 | 48.42 | 41.24 | 40.96 |
| **TIME** (Ours) | **94.92** | **93.59** | 83.29 | **100.00** | **98.93** | **94.15** |
| - w/o Selection | 90.35 | 88.52 | 74.82 | 100.00 | 97.14 | 90.17 |
| - w/o Mixing | 87.02 | 81.69 | 70.51 | 98.85 | 95.28 | 86.67 |
| - w/o $\mathcal{L}_{align}$ | 93.85 | 91.26 | 82.06 | 100.00 | 98.10 | 93.05 |
| - w/o Tensor | 94.48 | 90.87 | **83.88** | 100.00 | 98.22 | 93.49 |

LiveEdit (92.76 and 79.92 respectively) while reducing storage costs (as detailed in RQ2). This consistency verifies that TIME is a model-agnostic framework capable of generalizing to diverse VLMs without relying on model-specific engineering. More results are shown in Appendix F.

### 4.3. Parameter Efficiency and Inference Latency (RQ2)

**RQ2** investigates the scalability of TIME. We compare the storage cost and inference latency with the strongest baseline, LiveEdit, in Table 2.

**Memory Efficiency.** TIME drastically reduces the parameter footprint. For LLaVA-v1.5, TIME requires only 786 MB compared to LiveEdit's 1178 MB, achieving a reduction of approximately 33%. Similarly, on BLIP2, TIME reduces storage from 668 MB to 523 MB. This reduction stems from two factors: (1) the $O(R\sqrt{H})$ complexity of CP decomposition versus $O(rH)$ of LoRA, which is proved in Appendix D.3; and (2) the elimination of external routing modules. As shown in Figure 9 in Appendix F.2, as the hidden dimension $H$ grows, TIME's parameter growth is much slower than LiveEdit's, highlighting its advantage in

addressing the scalability bottleneck.

**Faster Inference.** By removing the auxiliary feature extractors and routing keys, TIME simplifies the inference path. Table 2 shows that TIME reduces editing time by approximately 32% on BLIP2 (e.g., 163.43s vs. 239.25s for single editing). Notably, in the lifelong setting (Gap=1000), this advantage expands, with TIME requiring only 178.92s compared to LiveEdit's 283.73s. This confirms that our intrinsic routing mechanism not only saves space but also accelerates execution by eliminating architectural redundancy.

### 4.4. Effectiveness of Intrinsic Components (RQ3)

To address **RQ3**, we perform comprehensive ablation studies on the E-VQA dataset (Table 3) by removing key components of TIME:

- **w/o Selection (Intrinsic Selection):** Replacing sparse selection with full retrieval leads to a significant drop in average score (94.15 → 90.17). This confirms that

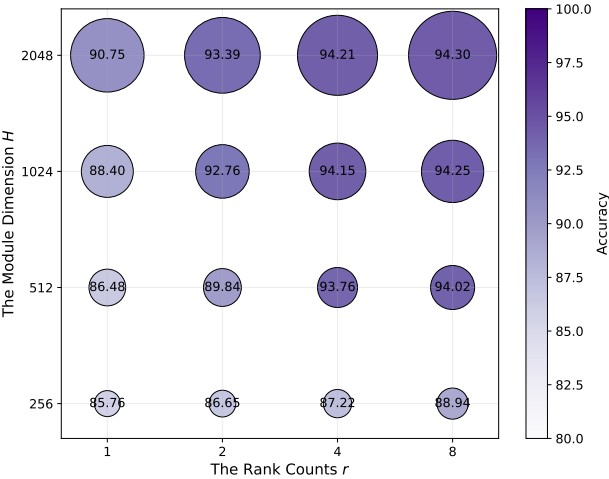

*Figure 5.* The impact of module dimension and rank counts on TIME performance. Results are lifelong editing (gaps=1000) on the E-VQA for BLIP2. Circle size represents training parameters, and color intensity represents average score.

the input factors of our CP tensors successfully define an *active subspace*, and sparse activation is crucial for preventing noise from irrelevant experts.

- **w/o Mixing (Intrinsic Scoring):** Removing the projection-score-based weighting causes the most severe degradation to 86.67. This indicates that the magnitude of the projection acts as a reliable confidence metric, ensuring that more relevant experts contribute more to the final prediction.

- **w/o $\mathcal{L}_{align}$ (Subspace Alignment):** Removing the structure-aware alignment objective results in a noticeable performance decline to 93.05. It suggests that without explicit supervision, the intrinsic projection scores may not spontaneously align perfectly with expert relevance. The alignment loss effectively forces the input factors to learn a discriminative subspace that accurately distinguishes between target and irrelevant queries without relying on external routers.

- **w/o Tensor (CP Decomposition):** Using full-rank matrices instead of CP decomposition results in a score of 93.49, which is slightly lower than our CP-based method. While full-rank matrices theoretically possess higher capacity, they are more prone to overfitting in few-shot editing scenarios. This demonstrates that CP decomposition enhances generalization while reducing parameter costs.

More ablation results are shown in Appendix F.3.

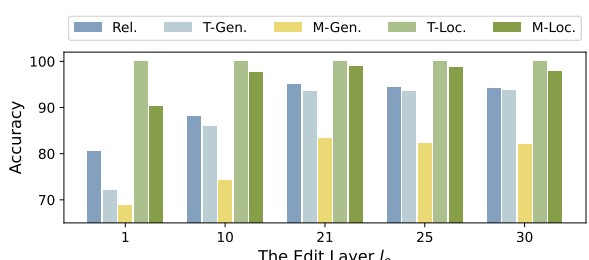

*Figure 6.* Impact of target layer index $l_e$ on E-VQA for BLIP2.

### 4.5. Impact of Hyperparameters (RQ4)

Finally, for **RQ4**, we analyze the sensitivity of TIME to structural hyperparameters.

**Module Dimension and Rank.** Figure 5 illustrates a grid search over module dimension $H$ and rank $R$. We observe a positive correlation between model capacity and performance. Notably, increasing $H$ from 256 to 2048 consistently improves the average score. However, larger dimensions increase parameter size. We choose $H = 1024, R = 4$ as a default to balance performance and efficiency.

**Target Layer Selection.** Figure 6 shows the impact of the editing layer index $l_e$. Performance is generally lower in shallow layers (e.g., layer 1-10) but stabilizes at a high level in deeper layers (21-30). This suggests that deeper layers in VLMs contain more semantic-rich representations suitable for knowledge editing.

### 4.6. Qualitative Analysis

We present detailed case studies and visual comparisons in Appendix F.1. As illustrated in Figure 7 and Figure 8, TIME demonstrates superior capability in generating semantically consistent and precise edits compared to baselines (LiveEdit). Specifically, our method effectively corrects multimodal hallucinations while preserving the integrity of unrelated visual-textual context.

## 5. Conclusion & Limitations

In this paper, we addressed the scalability bottleneck in lifelong multimodal knowledge editing, where traditional modular approaches suffer from prohibitive memory growth and architectural redundancy. We introduced TIME, a unified framework that harmonizes parameter efficiency with structural self-routing. By parameterizing edits as compact CP tensors and proposing a novel intrinsic routing mechanism, TIME utilizes the expert's active subspace to determine relevance directly, thereby eliminating the need for decoupled external retrievers. Extensive experiments demonstrate that TIME not only outperforms existing methods in editing sta-

bility and accuracy but also significantly reduces memory usage and inference latency. These results validate the potential of tensor factorization as a scalable foundation for next-generation lifelong knowledge editing systems.

**Limitations and Future Work.** TIME operates within two core trade-offs. First, its low-rank CP decomposition trades some expressivity for high efficiency, which is a fundamental constraint of compressed models. Our experiments (Sec. 4.5) confirm that the chosen rank captures targeted updates effectively, and the rank can be increased if more complex updates are needed. Second, factorized optimization differs from standard fine-tuning, but our alignment objective stabilizes training as shown in ablation studies (Sec. 4.4). Future work includes dynamic rank adaptation and applying TIME to broader multimodal tasks.

## Impact Statement

This paper presents work whose goal is to advance the field of multimodal knowledge editing, specifically improving the efficiency and stability of updating Vision-Language Models. Our method promotes the development of more trustworthy and sustainable AI systems by addressing model hallucinations and reducing retraining costs. While we do not foresee immediate negative societal consequences, we recognize the general dual-use nature of model editing technologies and encourage the community to develop accompanying safety verification protocols to prevent potential misuse.

## Acknowledgements

This work was supported by the Science and Technology Project of China Southern Power Grid Co., Ltd. (ZBKJXM20220010).

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

# A. Related Work

## A.1. Vision Language Models

Early vision language models (VLMs) established that large-scale cross-modal pretraining can yield strong alignment and broad transfer. Contrastive encoders such as CLIP (Radford et al., 2021) and ALIGN (Jia et al., 2021) aligned image–text pairs into a shared space, enabling open-vocabulary recognition and retrieval. Generative VLM families, e.g., BLIP (Li et al., 2022) and BLIP-2 (Li et al., 2023), further improved grounding by refining the visual interface and training curriculum. Building on these foundations, instruction-tuned multimodal LLMs (e.g., Flamingo (Alayrac et al., 2022), Kosmos-1 (Huang et al., 2023a), LLaVA (Liu et al., 2023), and Qwen-VL (Bai et al., 2025)) unified perception and language reasoning via visual tokens and instruction data.

Recent years have pushed VLMs along richer visual context and interleaving (Li et al., 2024a;b; Chen et al., 2024b) and more principled recipes and stronger encoders (Laurençon et al., 2024; Yao et al., 2025). Despite rapid progress, most VLMs remain static after training: updating a small set of visual/factual associations still requires retraining or global parameter changes. TIME directly targets this limitation by representing each edit as a compact CP-decomposed expert attached to a frozen backbone, enabling reliable updates without architectural bloat.

## A.2. Parameter Efficient Fine-tuning

Parameter Efficient Fine-tuning (PEFT) adapts large models by training lightweight modules while freezing the backbone. Prompt-based methods (prefix-tuning (Li & Liang, 2021), prompt-tuning (Lester et al., 2021), P-Tuning v2 (Liu et al., 2022b)) steer models through learned soft tokens. Residual-path approaches, including Adapters (Houlsby et al., 2019), LoRA (Hu et al., 2022), AdaLoRA (Zhang et al., 2023), and IA$^3$ (Liu et al., 2022a), inject compact updates via low-rank or feature-wise transformations. More recently, DoRA (Liu et al., 2024) decomposes weights into magnitude and direction, improving LoRA's stability and capacity without inference overhead. From a theoretical and optimization perspective, RepLoRA (Truong et al., 2025) revisits LoRA through an MoE lens and shows that simple reparameterizations can accelerate low-rank estimation.

For multimodal models, tensorized/decomposed updates offer an additional compression dimension. Prior work explored CP decomposition for efficient VLM adaptation (Wang et al., 2023), and LoRTA (Hounie et al., 2024) uses higher-order CP factorization to unify updates across layers matrices. However, standard PEFT is typically designed for single-task adaptation: in lifelong settings, stacking many modules causes parameter growth and requires separate routing/selection mechanisms. TIME diverges by storing edits as highly compact CP-factorized experts while using the factors themselves for intrinsic routing, eliminating separate selection parameters and achieving higher compression.

## A.3. Mixture-of-Experts and Routing Mechanism

Mixture-of-Experts (MoE) has re-emerged as a dominant scaling strategy due to sparse activation. Recent open models such as Mixtral (Jiang et al., 2024a), DeepSeekMoE (Dai et al., 2024), and DeepSeek-V2 (DeepSeek-AI et al., 2024) demonstrate that top-$k$ routing with expert specialization can approach dense-model quality with far lower active compute, while technical reports (e.g., Qwen2) further popularize MoE within broad model families (Yang et al., 2024). Alongside system advances, OpenMoE (Xue et al., 2024) provides an in-depth routing analysis, reporting that expert assignments can become context-independent and stabilize early in training, revealing inherent limitations of learned routers under scale.

Routing research also explores test-time re-routing (Li et al., 2025), self-selection (Lv et al., 2025), modality-aware experts (e.g., MoMa (Lin et al., 2024), SR-MoE (Delibasoglu, 2026)), and soft routing strategies for balancing multimodal fusion with language robustness (Xia et al., 2025). Addressing temporal dynamics, TiMoE (Faro et al., 2025) and Hierarchical Time-Aware MoE (Zhang et al., 2025b) integrate time-step information into gating for sequence modeling and recommendation, respectively, with Han et al. (2025) further guiding routing via temporal multimodal interactions. In contrast to auxiliary routers or iterative re-routing, TIME introduces intrinsic routing: the CP-decomposed expert defines an active subspace that deterministically matches inputs to edits. This design is particularly suitable for lifelong editing, where many small patches must coexist with minimal overhead and minimal interference.

## A.4. Knowledge Editing

Knowledge editing aims to modify localized knowledge while preserving unrelated behavior. For LLMs, direct weight interventions or learned update mappings (ROME (Meng et al., 2022), MEMIT (Meng et al., 2023), MEND (Mitchell et al., 2022a), SERAC (Mitchell et al., 2022b)) can reliably adjust specific facts but face scalability and interference as edits accumulate. Lifelong editing methods mitigate forgetting via retrieval or modular memories (LTE (Jiang et al., 2024b), RECIPE (Chen et al., 2024a), LEMoE (Wang & Li, 2024), MoKE (Cheng et al., 2025)). Importantly, WikiBigEdit (Thede et al., 2025) studies lifelong editing at practically relevant scale (hundreds of thousands of real-world Wikidata updates), highlighting that many editors degrade when pushed beyond synthetic small-scale benchmarks.

Extending editing to VLMs adds the challenge of maintaining visual–text alignment and cross-modal consistency. Recent benchmarks make these failure modes explicit: VLKEB (Huang et al., 2024) targets LVLM knowledge editing with improved data quality and portability evaluation, MMKE-Bench (Du et al., 2025) evaluates diverse real-world multimodal edits using free-form language and sequential settings, and MedMKEB (Xu et al., 2026) provides a benchmark for multimodal knowledge editing in the medical domain. Methodologically, UniKE (Pan et al., 2024) represents both internal and external knowledge as vectorized key-value pairs, enabling multimodal editing through semantic retrieval. CARML (Zhang et al., 2025a) achieves excellent editing results through external knowledge retrieval and multi-level knowledge guidance. LiveEdit (Chen et al., 2025) adopts low-rank experts with filtering and soft routing for lifelong VLM editing, but still relies on decoupled selection mechanisms for routing. MemEIC (Seong et al., 2026) studies continual and compositional knowledge editing for LVLMs, combining dual external memory and modality-specific LoRA adapters to preserve sequential edits while enabling cross-modal compositional reasoning.

However, these methods suffer from a scalability bottleneck due to their decoupled design, which relies on auxiliary semantic retrievers for expert selection. Building on this, our TIME framework introduces intrinsic routing, where the routing decision is induced directly from the expert's active subspace. By removing external retrievers and employing CP-decomposed experts, TIME achieves superior memory efficiency and state-of-the-art reliability in lifelong editing scenarios.

# B. Metrics and Editing Loss

## B.1. Metrics for Knowledge Editing

At timestep $t$, the updated model $f_{\theta_t}$ is evaluated along three standard dimensions of knowledge editing: **reliability**, **generality**, and **locality** (Cheng et al., 2023). We denote the corresponding metrics by $\mathcal{M}_{\mathrm{rel}}$, $\mathcal{M}_{\mathrm{gen}}$, and $\mathcal{M}_{\mathrm{loc}}$, following the multimodal variants used in existing editing benchmarks (e.g., Rel., T-Gen./I-Gen., T-Loc./I-Loc).

### B.1.1. RELIABILITY

Let $\mathcal{D}_t^{\mathrm{edit}} = \{(v_e, p_e, y_e)\}$ be the set of edit instances observed up to step $t$. Reliability measures the success rate of applying edits to their target pairs:

$$\mathcal{M}_{\mathrm{rel}}(f_{\theta_t}) = \mathbb{E}_{(v_e, p_e, y_e) \in \mathcal{D}_t^{\mathrm{edit}}} \Big[ \mathbf{1}\big\{ f_{\theta_t}(v_e, p_e) = y_e \big\} \Big]. \tag{23}$$

A higher $\mathcal{M}_{\mathrm{rel}}$ indicates more accurate incorporation of edited knowledge.

### B.1.2. GENERALITY

For each edit $e = (v_e, p_e, y_e)$, we consider semantically related neighbors to test whether the edit generalizes beyond the exact training instance.

Let $G_{\mathrm{text}}(e)$ denote a distribution over textual neighbors (e.g., rephrased prompts) paired with the same image $v_e$, and $G_{\mathrm{img}}(e)$ a distribution over visual neighbors (e.g., similar or augmented images) paired with the same prompt $p_e$. Then we can use **Text Generality** and **Image Generality** to evaluate knowledge editing methods:

$$\mathcal{M}_{\mathrm{gen}}^{\mathrm{text}}(f_{\theta_t}) = \mathbb{E}_{(v,p) \sim G_{\mathrm{text}}(e)} \Big[ \mathbf{1}\big\{ f_{\theta_t}(v, p) = y_e \big\} \Big]. \tag{24}$$

$$\mathcal{M}_{\mathrm{gen}}^{\mathrm{img}}(f_{\theta_t}) = \mathbb{E}_{(v,p) \sim G_{\mathrm{img}}(e)} \Big[ \mathbf{1}\big\{ f_{\theta_t}(v, p) = y_e \big\} \Big]. \tag{25}$$

High $\mathcal{M}_{\mathrm{gen}}^{\mathrm{text}}$ and $\mathcal{M}_{\mathrm{gen}}^{\mathrm{img}}$ indicate that the injected knowledge remains valid under reasonable multimodal variations.

### B.1.3. LOCALITY

Locality evaluates whether $f_{\theta_t}$ preserves its behavior on inputs that are unrelated to any edit.

Let $\mathcal{D}_{\text{text}}^{\text{loc}}$ and $\mathcal{D}_{\text{img}}^{\text{loc}}$ be distributions of locality test samples constructed to avoid semantic overlap with edited facts. For these samples, we measure consistency via prediction matching, either with the original model $f_{\theta_0}$ or with their gold labels when available. Then we can use **Text Locality** and **Image Locality** to evaluate knowledge editing methods:

$$\mathcal{M}_{\text{loc}}^{\text{text}}(f_{\theta_t}) = \mathbb{E}_{(v,p)\sim\mathcal{D}_{\text{text}}^{\text{loc}}} \left[ \mathbf{1}\left\{ f_{\theta_t}(v,p) = f_{\theta_0}(v,p) \right\} \right]. \tag{26}$$

$$\mathcal{M}_{\text{loc}}^{\text{img}}(f_{\theta_t}) = \mathbb{E}_{(v,p)\sim\mathcal{D}_{\text{img}}^{\text{loc}}} \left[ \mathbf{1}\left\{ f_{\theta_t}(v,p) = f_{\theta_0}(v,p) \right\} \right]. \tag{27}$$

Higher $\mathcal{M}_{\text{loc}}^{\text{text}}$ and $\mathcal{M}_{\text{loc}}^{\text{img}}$ reflect stronger preservation of unedited knowledge and reduced side effects. Together, these metrics provide a unified view of how well an editor achieves reliable updates, robust multimodal generalization, and minimal interference under lifelong editing.

### B.2. Details of Editing Loss

**Reliability Loss.** To ensure reliable editing, we supervise the output distribution of the injected representation $\mathbf{z}'$, using the edit sample set $\mathcal{D}_{\text{edit}} = \{(v_e, p_e, o_e)\}$ as the supervision target. Specifically, the loss is computed as:

$$\mathcal{L}_{\text{rel}} = \mathbb{E}_{(v_e, p_e, o_e)} \left[ -\log p_{\theta'}(o_e \mid v_e, p_e) \right], \tag{28}$$

where $\theta'$ represents the parameters of the model after editing.

**Generality Loss.** To improve semantic generalization, we define the paraphrasing or nearest-neighbor set $\mathcal{D}_g(e)$ for each edit $e$. The generality loss is given by:

$$\mathcal{L}_{\text{gen}} = \mathbb{E}_e \, \mathbb{E}_{(v,p)\in\mathcal{D}_g(e)} \left[ -\log p_{\theta'}(o_e \mid v, p) \right]. \tag{29}$$

This loss encourages the model to handle different semantic expressions for the same input effectively.

**Locality Loss.** To ensure that the edit does not affect unrelated samples too much, we constrain the distributions before and after injection to be close. For irrelevant samples $D_l = \{(v_l, p_l)\}$, we compute the per-token KL divergence, and the locality preservation loss is defined as:

$$\mathcal{L}_{\text{loc}} = \mathbb{E}_{(v_l, p_l)} \left[ \text{KL}\left( p_\theta(\cdot \mid v_l, p_l) \,\|\, p_{\theta'}(\cdot \mid v_l, p_l) \right) \right]. \tag{30}$$

This loss minimizes the difference between the pre-injection and post-injection distributions.

## C. Implementation Details and Algorithm

### C.1. More Details about Hyperparameters

The default hyperparameter settings of TIME are shown in Table 4. Training loss weights $\lambda_{align} = 0.5$, $\lambda_{rel} = \lambda_{gen} = \lambda_{loc} = 1$, and selection threshold $\gamma = 0.5$.

### C.2. Algorithm of TIME

The inference process of TIME with intrinsic routing is shown in Algorithm 1.

*Table 4.* Training hyperparameters of FT-L, FT-V and TIME.

| Editors | Backbones | Edit Iterations | Optimizer | Learning Rate | Edit Layers |
|---------|-----------|-----------------|-----------|---------------|-------------|
| FT-M | BLIP2 (2.7B) | 25 | AdamW | 1e-3 | The last layer of the language transformer |
|  | LLaVA (7B) | 25 | AdamW | 1e-3 | The last layer of the language transformer |
| FT-V | BLIP2 (2.7B) | 25 | AdamW | 1e-3 | Qformer |
|  | LLaVA (7B) | 25 | AdamW | 1e-3 | The multimodal projector |
| TIME | BLIP2 (2.7B) | 20 | AdamW | 1e-5 | layer 21 of the language transformer |
|  | LLaVA (7B) | 30 | AdamW | 1e-5 | layer 21 of the language transformer |

## D. Theoretical Analysis

### D.1. Equivalence to Eq.(14).

Let $\tilde{\mathbf{z}} = \text{vec}(Z)$. For the $k$-th rank channel:

$$
\begin{aligned}
\left[\widehat{U}\,\tilde{\mathbf{z}}\right]_k &= \tfrac{1}{\alpha\sqrt{R}}\,\mathbf{w}_k^{\text{in}\top}\,\text{vec}(Z) \\
&= \tfrac{1}{\alpha\sqrt{R}}\,(v_k^{\text{in}}\otimes u_k^{\text{in}})^\top\,\text{vec}(Z) \\
&= \tfrac{1}{\alpha\sqrt{R}}\,u_k^{\text{in}\top}\,Z\,v_k^{\text{in}}.
\end{aligned}
\tag{31}
$$

Then we can obtain:

$$
\sigma([\widehat{U}\,\tilde{\mathbf{z}}]_k) = \frac{1}{\alpha\sqrt{R}}\,s_k(Z).
\tag{32}
$$

Multiplying by $\widehat{V}^\top$ yields:

$$
\begin{aligned}
\mathbf{r}_m(\tilde{\mathbf{z}}) &= \sum_{k=1}^{R}\left(\tfrac{1}{\alpha\sqrt{R}}\,s_k(Z)\right)\tfrac{1}{\alpha\sqrt{R}}\,\mathbf{w}_k^{\text{out}} \\
&= \text{vec}\left(\tfrac{1}{\alpha\sqrt{R}}\sum_{k=1}^{R}s_k(Z)\,u_k^{\text{out}}v_k^{\text{out}\top}\right),
\end{aligned}
\tag{33}
$$

which exactly recovers the matrix-form output in Eq. (14). Therefore, Eq. (16) is a compact vectorized rewrite of Eq. (14).

### D.2. Proof of Convex Combination Stability for Eq. (20)

**Theorem D.1** (Convexity and Lipschitz Stability of Eq. (20)). *Let $\{r_m\}_{m\in\mathcal{M}^+}$ be residual operators satisfying $\|r_m(\tilde{z})\| \leq B$ and $\|r_m(\tilde{z}_1) - r_m(\tilde{z}_2)\| \leq L_m\|\tilde{z}_1 - \tilde{z}_2\|$ for every $m \in \mathcal{M}^+$, $\tilde{z}, \tilde{z}_1, \tilde{z}_2 \in \mathbb{R}^H$. Then the combined operator $r(\cdot)$ in (Eq. (20)) satisfies:*

(1) *(Convexity) $\pi_m \geq 0$ and $\sum_m \pi_m = 1$.*

(2) *(Boundedness) $\|r(\tilde{z})\| \leq B$.*

(3) *(Lipschitz Stability) $\|r(\tilde{z}_1) - r(\tilde{z}_2)\| \leq \left(\sum_m \pi_m L_m\right)\|\tilde{z}_1 - \tilde{z}_2\| \leq L_{\max}\|\tilde{z}_1 - \tilde{z}_2\|$, where $L_{\max} = \max_m L_m$.*

*Proof of Theorem D.1.* **(1) Convexity**: By construction, $g_m > 0$ and $\alpha_m > 0$, and softmax ensures $\sum_m \alpha_m = 1$. Since $\pi_m$ is the normalized product $g_m\alpha_m$, it is clear that $\pi_m \geq 0$ and $\sum_m \pi_m = 1$.

**(2) Boundedness**: For any vector norm, we have

$$
\begin{aligned}
\|r(\tilde{z})\| = \left\|\sum_m \pi_m r_m(\tilde{z})\right\| &\leq \sum_m \pi_m\|r_m(\tilde{z})\| \\
&\leq \sum_m \pi_m B = B.
\end{aligned}
\tag{34}
$$

---

**Algorithm 1** Inference Process of TIME with Intrinsic Routing

---

**Input:** Input hidden state $\mathbf{z} \in \mathbb{R}^H$, Layer Norm $\text{LN}(\cdot)$, Expert Repository $\mathcal{E} = \{\Theta_m\}_{m=1}^M$, Threshold $\gamma$, Temperature $\tau$.
**Output:** Edited hidden state $\mathbf{z}'$.

 1: **Step 1: Preprocessing**
 2: $\tilde{\mathbf{z}} \leftarrow \text{LN}(\mathbf{z})$
 3: $Z \leftarrow \text{reshape}(\tilde{\mathbf{z}}) \in \mathbb{R}^{s_1 \times s_2}$   {Reshape vector to matrix for tensor operation}
 4: **Step 2: Intrinsic Activation Scoring**
 5: Initialize scores $\mathcal{S} \leftarrow \{0\}^M$
 6: **for** each expert $m \in \{1, \ldots, M\}$ **do**
 7:   Extract input factors $\Theta_m^{\text{in}} = \{(u_{m,k}^{\text{in}}, v_{m,k}^{\text{in}})\}_{k=1}^R$ from $\mathcal{E}$
 8:   $S_m \leftarrow \sum_{k=1}^R \left| \sigma\left((u_{m,k}^{\text{in}})^\top Z\, v_{m,k}^{\text{in}}\right) \right|$   {Project input onto active subspace}
 9:   $\mathcal{S}[m] \leftarrow S_m$
10: **end for**
11: **Step 3: Sparse Selection & Weighting**
12: $\mathcal{M}_{\text{top}} \leftarrow \{m \mid S_m > \gamma\}$   {Select experts exceeding threshold $\gamma$}
13: **for** $m \in \mathcal{M}_{\text{top}}$ **do**
14:   $\pi_m \leftarrow \frac{\exp(S_m/\tau)}{\sum_{j \in \mathcal{M}_{\text{top}}} \exp(S_j/\tau)}$   {Input-dependent soft mixing}
15: **end for**
16: **Step 4: Expert Execution & Injection**
17: Initialize total residual $\mathbf{r}_{\text{final}} \leftarrow \mathbf{0}$
18: **for** $m \in \mathcal{M}_{\text{top}}$ **do**
19:   Extract output factors $\Theta_m^{\text{out}} = \{(u_{m,k}^{\text{out}}, v_{m,k}^{\text{out}})\}_{k=1}^R$
20:   {Ideally computed via vectorized form}
21:   Compute expert residual $\mathbf{r}_m(\tilde{\mathbf{z}})$ using $\Theta_m^{\text{in}}$ and $\Theta_m^{\text{out}}$
22:   $\mathbf{r}_{\text{final}} \leftarrow \mathbf{r}_{\text{final}} + \pi_m \cdot \mathbf{r}_m(\tilde{\mathbf{z}})$
23: **end for**
24: **Step 5: Final Update**
25: $\mathbf{z}' \leftarrow \mathbf{z} + \mathbf{r}_{\text{final}}$
26: **return** $\mathbf{z}'$

---

**(3) Lipschitz Stability**: Let $\tilde{z}_1, \tilde{z}_2 \in \mathbb{R}^H$:

$$\|r(\tilde{z}_1) - r(\tilde{z}_2)\| \leq \sum_m \pi_m \|r_m(\tilde{z}_1) - r_m(\tilde{z}_2)\|$$
$$\leq \sum_m \pi_m L_m \|\tilde{z}_1 - \tilde{z}_2\|$$
$$\leq L_{\max} \|\tilde{z}_1 - \tilde{z}_2\|. \tag{35}$$

Thus, the combined operator $r(\cdot)$ is no more unstable than the worst single expert. □

This result establishes that the update in Eq. (20) preserves both boundedness and Lipschitz smoothness under convex expert mixing, contributing to injection stability during editing.

### D.3. Proof of Lower Memory Complexity of TIME over LoRA+MoE

We compare the parameter count of a single expert under the same target layer of dimension $H \times H$, where $H$ is the hidden size of the edited layer.

**Baseline: LoRA+MoE Expert.** In standard modular editing approaches (e.g., LiveEdit), each expert is instantiated as a low-rank adapter (LoRA). It is parameterized by two matrices $A \in \mathbb{R}^{H \times r}$ and $B \in \mathbb{R}^{H \times r}$, where $r$ is the rank. The total

parameter count per expert is:

$$P_{\text{LoRA}} = 2Hr. \tag{36}$$

Additionally, these methods typically require an external routing module (e.g., stored keys $\mathbf{K} \in \mathbb{R}^{M \times d}$ or a feature extractor), introducing extra parameters $P_{\text{router}} > 0$.

**Ours: TIME Expert.** In TIME, we reshape the edited layer into a 4-way tensor of shape $(s_1, s_2, s_1, s_2)$ with $H = s_1 s_2$. We apply CP decomposition with rank $R$:

$$\mathcal{T} \approx \sum_{k=1}^{R} u_k^{\text{in}} \circ v_k^{\text{in}} \circ u_k^{\text{out}} \circ v_k^{\text{out}}, \tag{37}$$

where $u_k^{\text{in}}, u_k^{\text{out}} \in \mathbb{R}^{s_1}$ and $v_k^{\text{in}}, v_k^{\text{out}} \in \mathbb{R}^{s_2}$. Crucially, due to our **Intrinsic Routing** mechanism, the input factors $(u^{\text{in}}, v^{\text{in}})$ simultaneously serve as the routing parameters. Thus, the total number of trainable parameters per expert is:

$$P_{\text{TIME}} = R(2s_1 + 2s_2). \tag{38}$$

Assuming a square reshaping $s_1 = s_2 = \sqrt{H}$, this simplifies to:

$$P_{\text{TIME}} = 4R\sqrt{H}. \tag{39}$$

**Theorem D.2** (Space Advantage of TIME). *Under an equal rank budget ($R = r$) and assuming typical hidden dimensions ($H \geq 4$), a single TIME expert requires strictly fewer parameters than a LoRA expert. The compression ratio scales with $\sqrt{H}$:*

$$\frac{P_{TIME}}{P_{LoRA}} = \frac{4r\sqrt{H}}{2Hr} = \frac{2}{\sqrt{H}}. \tag{40}$$

*Proof.* Comparing the two parameter counts:

$$P_{\text{TIME}} < P_{\text{LoRA}} \iff 4R\sqrt{H} < 2Hr. \tag{41}$$

Let $R = r$. The inequality becomes:

$$2\sqrt{H} < H \iff 4 < H. \tag{42}$$

Since hidden dimensions $H$ in LLMs (e.g., 4096) are far greater than 4, the condition holds. More generally, TIME is more efficient whenever the tensor rank satisfies:

$$R < \frac{r}{2}\sqrt{H}. \tag{43}$$

For example, if $H = 4096$ ($\sqrt{H} = 64$) and $r = 8$, TIME remains efficient as long as $R < 256$, which easily covers practical rank settings (e.g., $R = 8$ or 16). $\qquad\square$

**System-Level Savings.** When scaling to $M$ experts, the total storage advantage of TIME is twofold:

1. **Expert Compression:** The linear growth of the repository is significantly slower ($M \cdot 4R\sqrt{H} \ll M \cdot 2Hr$).

2. **Zero-Overhead Routing:** Unlike baselines where Total Params $\approx \sum P_{\text{expert}} + P_{\text{router}}$, TIME achieves $P_{\text{router}} = 0$.

This confirms the theoretical space efficiency observed in Table 2 of the main text.

## E. Details about Datasets and Baselines

In this section, we summarize the datasets, VLLM backbones, and baseline editors used in our experiments.

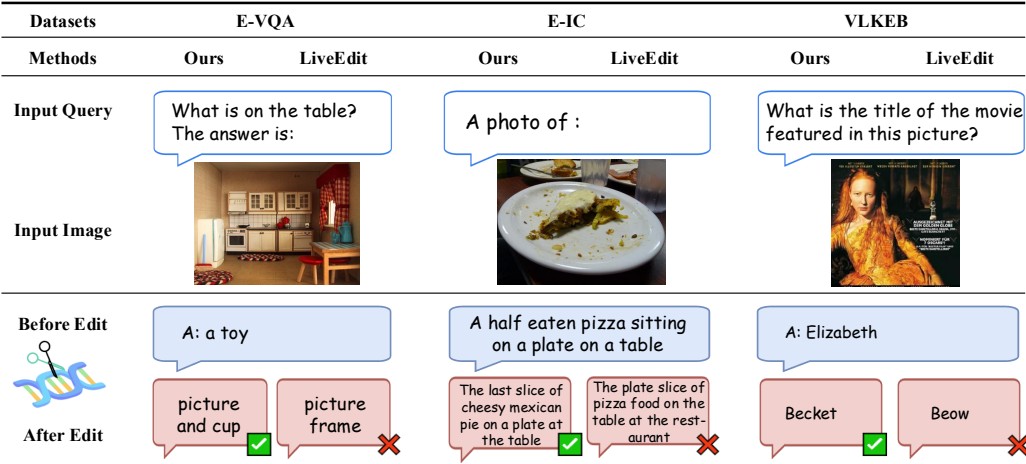

*Figure 7.* Case studies on three datasets. The backbone model is BLIP2 and the gap is 1.

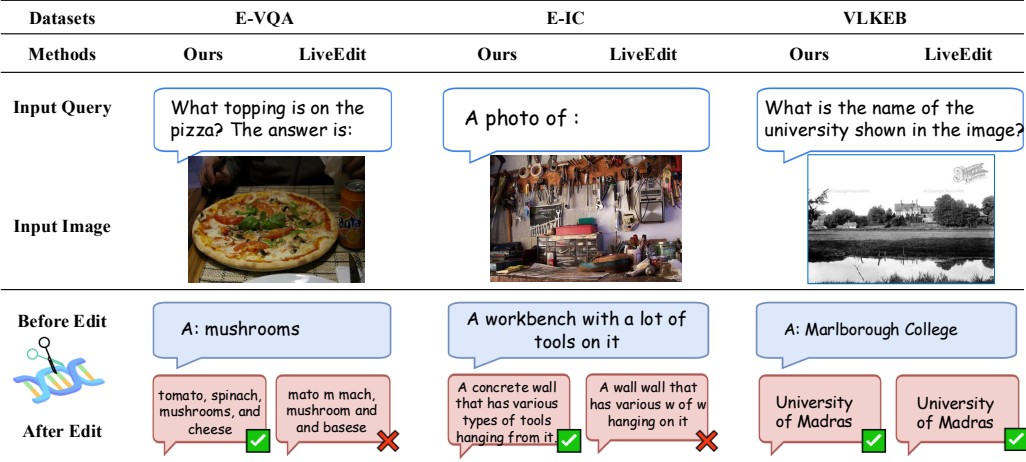

*Figure 8.* Case studies on three datasets. The backbone model is LLaVA-v1.5 and gap is 1000.

### E.1. Datasets

**E-VQA (Cheng et al., 2023):** We use an editing-oriented Visual Question Answering benchmark, where each instance specifies an image, a question, and a desired corrected answer. It evaluates whether an edited model can reliably produce the updated answer and generalize to semantically related queries under similar visual contexts.

**E-IC (Cheng et al., 2023):** This dataset contains image–caption pairs constructed for editing. Each edit defines a targeted change to the caption (e.g., entity, attribute, or relation), enabling us to test whether the model can update its descriptions while preserving consistency on unedited content.

**VLKEB (Huang et al., 2024):** VLKEB is a large-scale benchmark for knowledge editing in vision language models. It provides entity-centric image–text examples with annotated pre-edit and post-edit facts, supporting fine-grained evaluation of reliability, generality, and locality across diverse edits.

### E.2. Vision Language Model Backbones

We conduct experiments on three representative open-source VLMs, covering different architectures and training pipelines:

- **LLaVA (Liu et al., 2023):** An instruction-tuned VLM that connects a vision encoder to an LLM via a projection layer, widely used as a strong baseline for multimodal alignment.

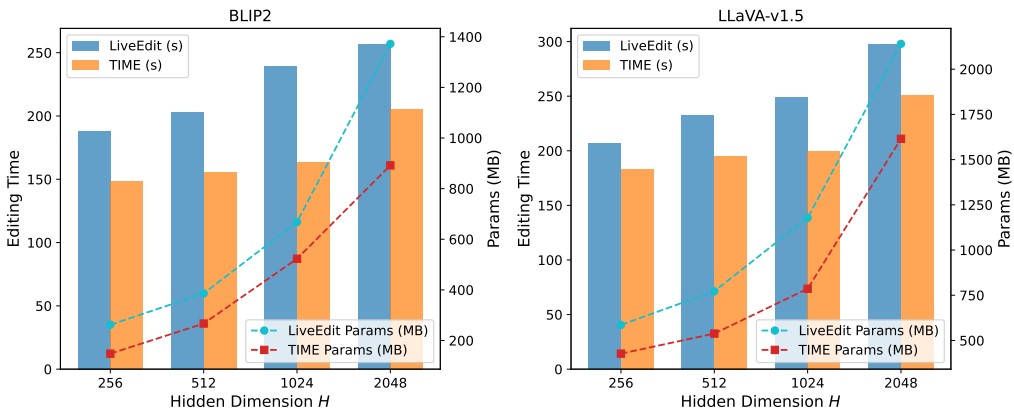

*Figure 9.* Comparison of editing time and editor parameters between LiveEdit and TIME on the E-VQA dataset. Editing time includes routing time.

- **BLIP-2** ([Li et al., 2023](#)): A Q-Former-based model that bridges a vision encoder and language model, providing a complementary architecture for studying editing behaviors.

For each backbone, we follow the standard publicly released checkpoints and inference settings unless otherwise specified.

### E.3. Baseline Editors

We compare our method with a comprehensive set of representative open-source editing approaches:

- **FT** (Fine-Tuning) directly fine-tunes model parameters on edit instances; serves as a simple but strong baseline.

- **MEND** ([Mitchell et al., 2022a](#)) (Model Editor Networks with Decomposition) learns a lightweight updater network that predicts low-rank parameter shifts conditioned on each edit.

- **SERAC** ([Mitchell et al., 2022b](#)) (Semi-parametric Editing with Retrieval-Augmented Counterfactuals) stores edit-specific memories and retrieves counterfactuals at inference to override original outputs.

- **TP** ([Huang et al., 2023b](#)) (Transformer Patcher) modifies intermediate hidden representations with small learned patches instead of updating full parameters.

- **LTE** ([Jiang et al., 2024b](#)) (Learning To Edit) trains a meta-editor to generate targeted parameter updates that generalize across edits.

- **RECIPE** ([Chen et al., 2024a](#)) (REtrieval-augmented ContInue Prompt lEarning) utilizes retrieval and continuous prompts to inject new knowledge without directly altering backbone weights.

- **LEMoE** ([Wang & Li, 2024](#)) (Lifelong Editing with Mixture of Experts) maintains an expandable mixture-of-experts for sequential edits, designed to improve lifelong reliability and locality.

- **LiveEdit** ([Chen et al., 2025](#)) is a recent editing framework that performs online, instance-wise updates guided by local supervision signals, enabling fast application of new edits while keeping changes compact and constrained to task-relevant regions of the model.

## F. More Experimental Results

### F.1. Qualitative Analysis

To qualitatively demonstrate the effectiveness of our proposed method, we present a comparative case study on three datasets against the baseline LiveEdit. As shown in Figure 7 and Figure 8, our method achieves more accurate and semantically

*Table 5.* More ablation results on E-VQA lifelong editing for BLIP2 (Gap=1000).

| Methods | Rel. | T-Gen. | I-Gen. | T-Loc. | I-Loc. | Avg. |
|---|---|---|---|---|---|---|
| Fine-tuning | 45.10 | 34.62 | 35.42 | 48.42 | 41.24 | 40.96 |
| LiveEdit | 94.42 | 91.98 | **84.65** | 100.00 | 97.38 | 93.65 |
| - with Tensor Factorization | 91.22 | 83.74 | 72.60 | 97.45 | 96.82 | 88.37 |
| - with Intrinsic Routing | 92.57 | 89.51 | 81.72 | 100.00 | 97.95 | 92.35 |
| **TIME** (Ours) | **94.92** | **93.59** | 83.29 | **100.00** | **98.93** | **94.15** |
| - w/o Selection | 90.35 | 88.52 | 74.82 | 100.00 | 97.14 | 90.17 |
| - w/o Mixing | 87.02 | 81.69 | 70.51 | 98.85 | 95.28 | 86.67 |
| - w/o $\mathcal{L}_{align}$ | 93.85 | 91.26 | 82.06 | 100.00 | 98.10 | 93.05 |
| - w/o Tensor Factorization | 94.48 | 90.87 | **83.88** | 100.00 | 98.22 | 93.49 |
| - with External Routing | 92.05 | 89.34 | 80.14 | 99.86 | 97.53 | 91.78 |

consistent edits in both visual and textual modalities. In the case of E-IC, our method effectively captures fine-grained visual semantics, revising the description to *"The last slice of cheesy mexican pie on a plate at the table"*, which aligns closely with the ground-truth depiction. However, LiveEdit generates verbose and inconsistent outputs, failing to maintain object–context coherence.

### F.2. More Analysis for Memory Cost and Edit Time

As illustrated in Figure 9, TIME consistently outperforms LiveEdit across both BLIP2 and LLaVA-v1.5 backbones in terms of editing efficiency and parameter economy. Specifically, TIME achieves markedly lower editing time under all hidden dimensions, demonstrating its superior scalability as model size increases. For instance, at $H = 2048$, TIME reduces the editing latency by approximately 20% compared with LiveEdit. More importantly, the parameter size of TIME grows much more slowly with increasing hidden dimension, confirming its parameter-efficient design. While LiveEdit expands from ∼300 MB to ∼1400 MB on BLIP2 and from ∼600 MB to over 2100 MB on LLaVA-v1.5, TIME maintains a compact representation. Specifically, on BLIP2, TIME requires only roughly 148 MB to 892 MB, and on LLaVA-v1.5, it ranges from approximately 427 MB to 1615 MB. These results indicate that TIME not only accelerates the editing process but also mitigates the problem of parameter explosion during lifelong editing, thereby offering a more practical and scalable solution for continual multimodal knowledge updates in VLMs.

### F.3. More Ablation Studies: Architectural Superiority Analysis

To demonstrate that the superiority of TIME stems from its **unified architectural design**, we conducted an in-depth "cross-architecture" ablation study on the E-VQA dataset (BLIP2, Gap=1000). Our objective is to verify whether the proposed CP-factorization and Intrinsic Routing are universally applicable modules, or if they require the specific holistic framework of TIME to function effectively.

**Experimental Settings.** We constructed two sets of structural variants to bridge the gap between the baseline LiveEdit and our TIME:

- **Grafting onto LiveEdit (Testing Baseline Limits):** We attempted to integrate our components into the decoupled architecture of LiveEdit:

  - *LiveEdit + Tensor Factorization:* Replacing LoRA experts with CP tensors while keeping the external retriever. This tests if tensor compression alone brings gains without intrinsic routing.
  - *LiveEdit + Intrinsic Routing:* Removing the external retriever and attempting to route based on LoRA parameter projections. This tests if intrinsic routing is feasible without the explicit factor structure of tensors.

- **Deconstructing TIME (Testing Unified Design):** We dismantled the unified structure of TIME to validate the necessity of its components:

 – *TIME with External Routing:* Disabling intrinsic routing and training an auxiliary external router (similar to LiveEdit) to select CP tensor experts.

**Analysis.** The results in Table 5 reveal the fundamental incompatibility of decoupled architectures with advanced editing mechanisms, confirming the unique advantage of TIME:

**1. Structural Incompatibility of Decoupled Baselines.** Simply grafting our components onto LiveEdit results in significant performance degradation.

- *LiveEdit + Tensor Factorization* drops drastically to 88.37. This indicates that CP-decomposed experts rely heavily on a specialized routing logic. LiveEdit's external retriever, which selects experts based on generic semantic features, fails to align with the fine-grained, structure-dependent active subspaces of tensor experts.

- *LiveEdit + Intrinsic Routing* (92.35) also fails to match the baseline. This confirms that Intrinsic Routing is not a standalone module; it requires the explicit input factors ($U^{in}$, $V^{in}$) of CP tensors to define a clean projection subspace. LoRA matrices lack this structural property, making them ill-suited for self-routing.

These failures prove that LiveEdit's **decoupled architecture** is a bottleneck: it cannot support the synergy between parameter-efficient structures and self-routing logic.

**2. The Necessity of Unified Design.** In contrast, TIME achieves its peak performance (94.15) only when both components work in unison. Breaking this unity by using *External Routing* causes a drop to 91.78, reinforcing that "Routing-Execution Consistency" is critical. Furthermore, *TIME w/o Tensor Factorization* (93.49) performs slightly worse than the CP-based version, showing that beyond efficiency, the CP structure provides beneficial regularization that full matrices lack.

In summary, TIME is not merely a combination of "LiveEdit + Tensor". It is a holistic framework where the storage format (CP Tensor) and the control logic (Intrinsic Routing) are mathematically coupled. This unified design is the key driver of its superior stability and efficiency.

### F.4. The Complete Lifelong Editing Results

Tables 6, 7, and 8 present the comprehensive results of lifelong editing on the E-VQA, E-IC, and VLKEB benchmarks, respectively. In scenarios with longer editing gaps, TIME demonstrated better performance than LiveEdit. These results must be interpreted within the context of the efficiency-performance trade-off:

**1. Breaking the Pareto Frontier.** It is crucial to emphasize that TIME achieves these performance gains while significantly reducing computational overhead. As detailed in Section 4.3, TIME operates with ∼33% fewer parameters and ∼32% faster inference speed compared to LiveEdit. Typically, such drastic compression leads to performance degradation. The fact that TIME not only matches but surpasses the baseline's accuracy under strict resource constraints demonstrates a superior Pareto efficiency. TIME effectively proves that massive, decoupled routing parameters are redundant for high-performance editing.

**2. Superiority without External Aid.** The comparison is structurally tilted against TIME: LiveEdit relies on an auxiliary, decoupled retriever (external keys) to explicitly guide expert selection, which can be viewed as additional semantic supervision. In contrast, TIME employs intrinsic routing, forcing the model to self-organize using only the active subspaces of the edit parameters. Outperforming a method equipped with an external navigator using a purely self-contained intrinsic mechanism highlights the fundamental robustness and distinctiveness of our proposed subspace modeling.

**3. Universal Robustness across Scenarios.** The extensive evaluation across three distinct datasets (E-VQA, E-IC, VLKEB) and two different model architectures (BLIP2, LLaVA) confirms the wide applicability of our method. Unlike baselines that may fluctuate in performance depending on the data distribution or model size, TIME maintains consistent superiority across all experimental settings. This universality validates that the proposed Intrinsic Routing is a fundamental improvement applicable to the general class of VLM editing problems, rather than a heuristic tuned for a specific dataset.

*Table 6.* A comparison with previous work on E-VQA. Best metrics are indicated by **bold**.

| Dataset | #Gap | Editor | BLIP2 | | | | | | LLaVA-v1.5 | | | | | |
|---|---|---|---|---|---|---|---|---|---|---|---|---|---|---|
| | | | Rel. | T-Gen. | I-Gen. | T-Loc. | I-Loc. | Avg. | Rel. | T-Gen. | I-Gen. | T-Loc. | I-Loc. | Avg. |
| E-VQA | 1 | FT-L | 52.86 | 48.80 | 32.94 | 98.24 | 94.27 | 65.42 | 93.88 | 87.98 | 80.25 | 99.61 | 94.78 | 91.30 |
| | | FT-V | 91.70 | 87.24 | 33.30 | **100.00** | 85.22 | 79.49 | 87.29 | 76.11 | 53.23 | **100.00** | 96.95 | 82.72 |
| | | MEND | 93.13 | 92.76 | 93.07 | 92.00 | 75.81 | 89.35 | 91.23 | 90.05 | 91.29 | 91.02 | 90.22 | 90.76 |
| | | SERAC | 88.39 | 84.50 | 84.25 | 85.82 | 26.00 | 73.79 | 89.33 | 83.72 | 84.97 | 82.05 | 23.78 | 72.77 |
| | | TP | 70.14 | 65.80 | 53.05 | 98.11 | 85.33 | 74.49 | 35.95 | 36.12 | 28.65 | 93.87 | 97.61 | 58.44 |
| | | LTE | 95.74 | 93.86 | 86.90 | 97.93 | 87.97 | 92.48 | 94.16 | 93.57 | **93.59** | 94.08 | 86.26 | 92.33 |
| | | RECIPE | 89.42 | 86.24 | 87.53 | 99.87 | 89.16 | 90.45 | 91.37 | 86.51 | 87.73 | 94.27 | 88.88 | 89.75 |
| | | LEMoE | 93.56 | 92.23 | 91.40 | 98.50 | 85.21 | 92.18 | 93.60 | 92.77 | 89.99 | 99.28 | 96.98 | 94.52 |
| | | LiveEdit | **96.67** | 94.20 | **93.82** | **100.00** | **100.00** | **96.94** | 94.28 | 94.51 | 88.01 | **100.00** | **100.00** | **95.36** |
| | | Ours | 96.55 | **94.65** | 92.34 | **100.00** | 99.98 | 96.70 | **94.62** | **94.75** | 86.80 | **100.00** | **100.00** | 95.23 |
| | 10 | FT-L | 53.77 | 48.91 | 37.66 | 97.68 | 80.40 | 63.68 | 90.57 | 84.14 | 73.21 | 95.56 | 81.50 | 85.00 |
| | | FT-V | 85.27 | 81.09 | 34.60 | **100.00** | 49.00 | 69.99 | 84.90 | 73.53 | 49.99 | **100.00** | 55.98 | 72.88 |
| | | MEND | 40.40 | 32.21 | 27.73 | 87.92 | 67.74 | 51.20 | 3.58 | 3.55 | 3.53 | 2.10 | 1.26 | 2.80 |
| | | SERAC | 86.12 | 82.24 | 82.52 | 70.24 | 16.06 | 67.43 | 88.09 | 83.40 | 83.57 | 64.91 | 15.50 | 67.10 |
| | | TP | 55.86 | 52.39 | 43.00 | 91.26 | 63.62 | 61.22 | 32.71 | 31.23 | 28.58 | 75.10 | 91.17 | 51.76 |
| | | LTE | 94.96 | 92.43 | 85.23 | 94.19 | 87.01 | 90.76 | 92.83 | 91.41 | **90.82** | 86.38 | 85.52 | 89.39 |
| | | RECIPE | 88.27 | 85.37 | 84.80 | 97.78 | 88.49 | 88.94 | 90.22 | 85.92 | 86.24 | 90.34 | 88.11 | 88.17 |
| | | LEMoE | 91.59 | 85.92 | 86.82 | 83.64 | 29.71 | 75.54 | 91.95 | 86.54 | 79.82 | 85.19 | 49.81 | 78.66 |
| | | LiveEdit | **95.65** | 93.46 | **90.98** | **100.00** | **100.00** | **96.02** | 93.79 | **93.21** | 86.42 | **100.00** | **100.00** | **94.68** |
| | | Ours | 95.50 | **94.38** | 89.53 | **100.00** | 99.95 | 95.87 | **94.02** | 93.18 | 85.94 | **100.00** | **100.00** | 94.63 |
| | 100 | FT-L | 52.47 | 43.68 | 39.43 | 91.13 | 47.72 | 54.88 | 79.67 | 70.05 | 64.07 | 83.47 | 54.44 | 70.34 |
| | | FT-V | 48.04 | 40.71 | 29.86 | **100.00** | 33.25 | 50.37 | 82.90 | 72.83 | 47.26 | **100.00** | 43.39 | 69.28 |
| | | MEND | 17.69 | 16.39 | 18.31 | 91.52 | 67.94 | 42.37 | 2.22 | 2.20 | 2.21 | 0.21 | 0.62 | 1.49 |
| | | SERAC | 86.98 | 81.27 | 80.97 | 71.04 | 15.55 | 67.17 | 88.08 | 81.53 | 82.48 | 62.13 | 12.90 | 65.42 |
| | | TP | 44.26 | 38.24 | 33.28 | 43.81 | 38.54 | 39.63 | 29.37 | 28.72 | 24.66 | 14.64 | 45.01 | 28.48 |
| | | LTE | 92.78 | 90.36 | 83.88 | 94.33 | 81.79 | 88.63 | 88.92 | 87.89 | 85.72 | 84.34 | 81.60 | 85.69 |
| | | RECIPE | 88.12 | 82.49 | 83.10 | 98.97 | 86.53 | 87.84 | 89.86 | 83.32 | 84.82 | 87.37 | 85.08 | 86.09 |
| | | LEMoE | 29.61 | 21.70 | 27.05 | 79.48 | 32.61 | 38.09 | 42.41 | 36.60 | 34.33 | 78.57 | 53.28 | 49.04 |
| | | LiveEdit | **95.25** | 92.94 | 85.48 | **100.00** | **99.76** | 94.69 | 93.54 | 92.34 | **85.89** | **100.00** | 99.31 | 94.21 |
| | | Ours | **95.25** | **93.90** | **86.85** | **100.00** | 99.54 | **95.11** | **93.86** | **92.65** | 85.41 | **100.00** | **99.58** | **94.30** |
| | 1000 | FT-L | 45.10 | 34.62 | 35.42 | 48.42 | 41.24 | 40.96 | 71.39 | 59.83 | 57.41 | 55.55 | 48.99 | 58.63 |
| | | FT-V | 40.40 | 31.46 | 27.85 | **100.00** | 27.44 | 45.43 | 69.57 | 56.34 | 44.07 | **100.00** | 41.47 | 62.29 |
| | | MEND | 15.84 | 14.35 | 17.73 | 91.74 | 70.17 | 41.97 | 0.04 | 0.05 | 0.05 | 0.08 | 0.09 | 0.06 |
| | | SERAC | 83.35 | 70.80 | 80.32 | 67.66 | 13.13 | 63.05 | 85.57 | 75.58 | 82.01 | 62.46 | 15.69 | 64.26 |
| | | TP | 20.63 | 15.09 | 18.41 | 8.65 | 8.25 | 14.21 | 16.56 | 16.80 | 15.65 | 7.28 | 15.60 | 14.38 |
| | | LTE | 89.32 | 82.82 | 81.51 | 94.86 | 69.83 | 83.67 | 83.93 | 82.55 | 81.34 | 83.97 | 73.09 | 80.98 |
| | | RECIPE | 84.99 | 74.20 | 82.04 | 96.82 | 87.73 | 85.16 | 87.00 | 76.81 | 83.09 | 86.95 | 87.03 | 84.18 |
| | | LEMoE | 19.73 | 17.34 | 18.22 | 72.01 | 31.06 | 31.67 | 30.80 | 25.75 | 24.32 | 71.45 | 46.23 | 39.71 |
| | | LiveEdit | 94.42 | 91.98 | **84.65** | **100.00** | 97.38 | 93.69 | 92.93 | 90.16 | **84.30** | **100.00** | 96.43 | 92.76 |
| | | Ours | **94.92** | **93.59** | 83.29 | **100.00** | 98.93 | **94.15** | **93.92** | **91.41** | 83.47 | **100.00** | **96.92** | **93.14** |

*Table 7.* A comparison with previous work on E-IC. Best metrics are indicated by **bold**.

| Dataset | #Gap | Editor | BLIP2 | | | | | | LLaVA-v1.5 | | | | | |
|---|---|---|---|---|---|---|---|---|---|---|---|---|---|---|
| | | | Rel. | T-Gen. | I-Gen. | T-Loc. | I-Loc. | Avg. | Rel. | T-Gen. | I-Gen. | T-Loc. | I-Loc. | Avg. |
| **E-IC** | 1 | FT-L | 45.02 | 44.47 | 40.72 | 99.02 | 98.27 | 65.50 | 73.48 | 72.98 | 65.79 | 99.28 | 99.06 | 82.12 |
| | | FT-V | 67.14 | 61.76 | 43.34 | **100.00** | 96.76 | 73.80 | 56.19 | 56.55 | 49.94 | **100.00** | **100.00** | 72.54 |
| | | MEND | **94.96** | **92.45** | **92.33** | 94.95 | 88.86 | 92.71 | 92.82 | **91.81** | 90.59 | 96.38 | 93.69 | **93.06** |
| | | SERAC | 88.71 | 83.81 | 84.38 | 84.28 | 24.70 | 73.18 | 88.18 | 81.03 | 85.61 | 84.01 | 28.58 | 73.48 |
| | | TP | 49.65 | 48.58 | 46.02 | 93.69 | 78.95 | 63.38 | 57.63 | 59.23 | 55.34 | 60.90 | 88.00 | 64.22 |
| | | LTE | 92.58 | 91.94 | 90.90 | 97.80 | 91.42 | **92.93** | 93.35 | 91.30 | **92.77** | 95.77 | 91.98 | 93.03 |
| | | RECIPE | 85.20 | 81.44 | 82.71 | **100.00** | 94.59 | 88.79 | 84.45 | 76.97 | 81.57 | 96.53 | 96.37 | 87.18 |
| | | LEMoE | 93.07 | 91.37 | 83.28 | 94.45 | 60.44 | 84.52 | **93.80** | 91.42 | 90.61 | 95.14 | 93.00 | 92.79 |
| | | LiveEdit | 80.60 | 80.12 | 76.88 | **100.00** | **100.00** | 87.52 | 82.16 | 81.01 | 78.27 | **100.00** | **100.00** | 88.29 |
| | | Ours | 82.21 | 80.44 | 75.82 | **100.00** | **100.00** | 87.69 | 81.23 | 82.92 | 80.33 | **100.00** | **100.00** | 88.90 |
| | 10 | FT-L | 46.20 | 44.07 | 40.54 | 98.37 | 92.59 | 64.35 | 68.74 | 67.05 | 60.95 | 97.05 | 91.20 | 77.00 |
| | | FT-V | 63.79 | 56.88 | 47.75 | **100.00** | 52.41 | 64.17 | 57.23 | 55.92 | 50.09 | **100.00** | 90.31 | 70.71 |
| | | MEND | 7.15 | 7.48 | 7.01 | 19.74 | 21.70 | 12.62 | 66.49 | 64.77 | 58.42 | 87.25 | 86.15 | 72.62 |
| | | SERAC | 47.69 | 46.94 | 43.46 | 65.29 | 16.65 | 44.01 | 56.57 | 56.88 | 52.62 | 59.96 | 14.66 | 48.14 |
| | | TP | 44.57 | 44.88 | 39.82 | 65.84 | 51.55 | 49.33 | 45.28 | 47.25 | 42.78 | 19.74 | 59.59 | 42.93 |
| | | LTE | 47.56 | 50.73 | 44.64 | 97.05 | 92.90 | 66.57 | 52.16 | 55.70 | 48.39 | 90.74 | 89.09 | 67.21 |
| | | RECIPE | 47.26 | 46.30 | 42.83 | **100.00** | 95.38 | 66.35 | 56.00 | 56.19 | 52.14 | 91.80 | 95.31 | 70.29 |
| | | LEMoE | **87.45** | **84.54** | 71.74 | 85.98 | 71.23 | 80.19 | **89.00** | **85.23** | **83.24** | 86.39 | 82.86 | 85.34 |
| | | LiveEdit | 80.65 | 79.38 | **75.66** | **100.00** | **100.00** | **87.14** | 81.93 | 80.80 | 75.55 | **100.00** | **100.00** | 87.66 |
| | | Ours | 80.15 | 79.29 | 74.80 | **100.00** | **100.00** | 86.85 | 81.01 | 81.44 | 76.19 | **100.00** | **100.00** | **87.73** |
| | 100 | FT-L | 50.81 | 48.70 | 42.79 | 90.12 | 52.09 | 56.90 | 65.08 | 60.90 | 58.46 | 86.82 | 89.19 | 72.09 |
| | | FT-V | 56.23 | 53.18 | 43.37 | **100.00** | 31.59 | 56.87 | 59.41 | 55.44 | 52.16 | **100.00** | 72.93 | 67.99 |
| | | MEND | 7.97 | 8.19 | 8.04 | 20.15 | 23.23 | 13.52 | 56.83 | 56.89 | 53.07 | 87.63 | 84.64 | 67.81 |
| | | SERAC | 43.55 | 42.16 | 39.09 | 57.77 | 15.84 | 39.68 | 53.35 | 53.70 | 49.41 | 48.04 | 17.25 | 44.35 |
| | | TP | 37.51 | 37.62 | 33.80 | 10.26 | 20.84 | 28.00 | 22.88 | 25.81 | 20.90 | 3.59 | 14.87 | 17.61 |
| | | LTE | 43.03 | 42.52 | 39.63 | 96.20 | 91.95 | 62.67 | 48.37 | 49.63 | 45.08 | 87.66 | 87.62 | 63.67 |
| | | RECIPE | 43.40 | 42.19 | 39.02 | 98.40 | 95.55 | 63.71 | 53.23 | 53.75 | 49.36 | 87.64 | 95.52 | 67.90 |
| | | LEMoE | 57.13 | 51.73 | 46.73 | 91.72 | 58.06 | 61.08 | 55.36 | 53.14 | 51.12 | 87.77 | 81.35 | 65.75 |
| | | LiveEdit | **79.18** | 77.44 | **74.12** | **100.00** | **100.00** | **86.15** | **80.81** | 78.77 | 63.52 | **100.00** | **100.00** | **84.62** |
| | | Ours | 79.14 | 77.50 | 73.67 | **100.00** | **100.00** | 86.06 | 79.02 | **78.94** | **63.80** | **100.00** | **100.00** | 84.35 |
| | 1000 | FT-L | 53.61 | 48.81 | 45.92 | 52.45 | 59.09 | 51.98 | 59.78 | 54.99 | 54.17 | 65.37 | 78.96 | 62.65 |
| | | FT-V | 48.24 | 45.55 | 43.03 | **100.00** | 23.76 | 52.12 | 49.21 | 47.75 | 43.81 | **100.00** | 35.14 | 55.18 |
| | | MEND | 6.54 | 6.51 | 6.50 | 13.52 | 20.38 | 10.69 | 54.39 | 54.14 | 50.99 | 83.87 | 80.60 | 64.80 |
| | | SERAC | 43.12 | 41.69 | 38.72 | 48.08 | 14.88 | 37.30 | 52.93 | 53.44 | 49.01 | 49.91 | 16.65 | 44.39 |
| | | TP | 26.03 | 26.31 | 24.85 | 4.11 | 11.77 | 18.62 | 10.28 | 13.14 | 9.75 | 1.71 | 4.45 | 7.87 |
| | | LTE | 44.52 | 43.56 | 41.29 | 96.45 | 90.86 | 63.33 | 48.83 | 49.96 | 45.68 | 85.17 | 86.41 | 63.21 |
| | | RECIPE | 43.02 | 41.63 | 38.59 | 99.68 | 92.96 | 63.18 | 53.11 | 53.48 | 48.99 | 87.93 | 94.84 | 67.67 |
| | | LEMoE | 43.46 | 43.34 | 37.69 | 93.27 | 67.52 | 57.06 | 34.50 | 31.38 | 28.14 | 82.09 | 75.88 | 50.40 |
| | | LiveEdit | 72.86 | 70.34 | **67.92** | **100.00** | **100.00** | 82.22 | 72.80 | 69.95 | 57.05 | **100.00** | **99.79** | 79.92 |
| | | Ours | **74.24** | **72.58** | 65.81 | **100.00** | **100.00** | **82.53** | **73.32** | **71.69** | **61.04** | **100.00** | 99.52 | **81.12** |

*Table 8.* A comparison with previous work on VLKEB. Best metrics are indicated by **bold**. Second best metrics are indicated by underline.

| Dataset | #Gap | Editor | BLIP2 | | | | | | LLaVA-v1.5 | | | | | |
|---|---|---|---|---|---|---|---|---|---|---|---|---|---|---|
| | | | Rel. | T-Gen. | I-Gen. | T-Loc. | I-Loc. | Avg. | Rel. | T-Gen. | I-Gen. | T-Loc. | I-Loc. | Avg. |
| VLKEB | 1 | FT-L | 54.31 | 54.27 | 54.08 | 98.40 | 94.37 | 71.09 | 94.29 | 87.00 | 92.22 | 91.16 | 91.37 | 91.21 |
| | | FT-V | 92.64 | 80.97 | 63.62 | **100.00** | 83.02 | 84.05 | 76.31 | 65.57 | 59.43 | **100.00** | 92.35 | 78.73 |
| | | MEND | 94.91 | 93.81 | 93.84 | 94.98 | 86.54 | 92.82 | 92.13 | 91.28 | 90.22 | 89.19 | 90.13 | 90.59 |
| | | SERAC | 87.95 | 84.67 | 85.20 | 68.10 | 17.75 | 68.73 | 89.77 | 89.11 | 87.92 | 66.68 | 14.20 | 69.54 |
| | | TP | 50.98 | 49.47 | 50.88 | 94.76 | 78.57 | 64.93 | 50.77 | 55.70 | 51.65 | 87.93 | 90.43 | 67.30 |
| | | LTE | 94.13 | 91.93 | 92.23 | 93.89 | 92.27 | 92.89 | 94.42 | 93.57 | 93.22 | 86.84 | 79.69 | 89.55 |
| | | RECIPE | 92.38 | 89.74 | 89.17 | 97.13 | 94.46 | 92.58 | 92.67 | 92.35 | 91.01 | 89.67 | 82.85 | 89.71 |
| | | LEMoE | 94.59 | 93.14 | 92.37 | 94.53 | 61.53 | 87.23 | 94.85 | 93.09 | 91.67 | 87.03 | 87.88 | 90.90 |
| | | LiveEdit | **98.77** | **98.08** | 94.89 | **100.00** | **100.00** | **98.35** | **96.43** | 95.22 | 93.72 | **100.00** | **100.00** | 97.08 |
| | | Ours | 98.62 | 97.69 | **95.19** | **100.00** | **100.00** | 98.30 | 95.82 | **95.93** | **93.98** | **100.00** | **100.00** | **97.15** |
| | 10 | FT-L | 55.81 | 55.55 | 55.29 | 93.37 | 75.45 | 67.09 | 88.05 | 85.32 | 85.23 | 74.53 | 85.74 | 83.77 |
| | | FT-V | 92.48 | 86.02 | 71.48 | **100.00** | 60.34 | 82.06 | 68.63 | 57.57 | 56.56 | **100.00** | 82.99 | 73.15 |
| | | MEND | 54.47 | 52.59 | 53.61 | 91.96 | 84.58 | 67.44 | 0.18 | 0.24 | 0.05 | 0.03 | 0.19 | 0.14 |
| | | SERAC | 78.43 | 67.72 | 75.12 | 56.21 | 14.54 | 58.40 | 81.55 | 74.49 | 80.24 | 54.71 | 13.15 | 60.83 |
| | | TP | 50.19 | 51.61 | 50.13 | 89.26 | 68.48 | 61.93 | 44.56 | 47.52 | 45.36 | 52.21 | 66.61 | 51.25 |
| | | LTE | 92.02 | 80.25 | 90.30 | 94.66 | 90.39 | 89.52 | 90.06 | 81.52 | 88.11 | 83.40 | 81.48 | 84.91 |
| | | RECIPE | 81.39 | 70.31 | 79.19 | 94.40 | 95.59 | 84.18 | 83.92 | 76.23 | 82.84 | 86.33 | 83.69 | 82.60 |
| | | LEMoE | 91.40 | 90.71 | 89.12 | 58.20 | 49.49 | 75.79 | 91.55 | 84.58 | 81.03 | 67.19 | 72.81 | 79.43 |
| | | LiveEdit | **98.57** | **97.92** | 94.63 | **100.00** | **100.00** | **98.22** | **95.54** | 94.52 | 91.25 | **100.00** | **100.00** | 96.26 |
| | | Ours | 98.55 | 97.63 | **94.85** | **100.00** | **100.00** | 98.21 | 94.96 | **95.18** | **91.72** | **100.00** | **100.00** | **96.37** |
| | 100 | FT-L | 57.26 | 56.23 | 56.73 | 86.19 | 68.87 | 65.06 | 75.41 | 73.67 | 74.16 | 70.01 | 82.05 | 75.06 |
| | | FT-V | 56.49 | 58.16 | 53.13 | **100.00** | 43.52 | 62.26 | 60.60 | 59.79 | 56.29 | **100.00** | 68.07 | 68.95 |
| | | MEND | 39.91 | 40.96 | 40.37 | 92.32 | 84.32 | 59.58 | 0.56 | 0.58 | 0.66 | 0.18 | 0.07 | 0.41 |
| | | SERAC | 66.73 | 53.69 | 64.37 | 59.18 | 17.86 | 52.37 | 72.25 | 62.43 | 70.68 | 53.73 | 13.69 | 54.56 |
| | | TP | 46.61 | 48.12 | 47.22 | 64.25 | 43.21 | 49.88 | 19.71 | 20.07 | 19.36 | 11.40 | 24.05 | 18.92 |
| | | LTE | 78.41 | 60.70 | 78.55 | 94.34 | 90.13 | 80.42 | 80.27 | 64.25 | 80.13 | 81.62 | 79.11 | 77.08 |
| | | RECIPE | 68.79 | 54.96 | 67.20 | 94.61 | 97.12 | 76.54 | 73.97 | 63.73 | 72.69 | 86.20 | 82.59 | 75.84 |
| | | LEMoE | 44.13 | 44.78 | 42.55 | 58.13 | 51.76 | 48.27 | 83.07 | 75.36 | 71.72 | 54.09 | 49.68 | 66.78 |
| | | LiveEdit | 98.20 | **97.67** | **93.96** | **100.00** | **100.00** | **97.97** | **94.56** | 90.65 | **89.56** | **100.00** | **100.00** | **94.95** |
| | | Ours | **98.24** | 97.08 | 93.12 | **100.00** | **100.00** | 97.69 | 93.70 | **92.54** | 88.26 | **100.00** | **100.00** | 94.90 |
| | 1000 | FT-L | 55.39 | 54.34 | 53.87 | 50.80 | 54.00 | 53.68 | 68.14 | 66.38 | 66.98 | 65.61 | 75.35 | 68.49 |
| | | FT-V | 47.03 | 49.68 | 46.99 | **100.00** | 41.41 | 57.02 | 53.41 | 48.80 | 43.16 | **100.00** | 57.03 | 60.48 |
| | | MEND | 37.22 | 38.03 | 37.19 | 91.49 | 84.10 | 57.61 | 0.03 | 0.05 | 0.07 | 0.06 | 0.08 | 0.06 |
| | | SERAC | 53.58 | 45.78 | 52.42 | 56.81 | 16.90 | 45.10 | 60.93 | 56.49 | 60.06 | 52.94 | 15.04 | 49.09 |
| | | TP | 24.36 | 24.21 | 24.25 | 16.37 | 19.96 | 21.83 | 5.46 | 4.81 | 5.51 | 2.77 | 7.19 | 5.15 |
| | | LTE | 61.67 | 51.05 | 61.60 | 94.78 | 90.94 | 72.01 | 64.51 | 56.26 | 64.80 | 80.85 | 76.52 | 68.59 |
| | | RECIPE | 54.64 | 46.54 | 54.10 | 94.60 | 96.93 | 69.37 | 62.00 | 56.84 | 61.50 | 85.37 | 82.07 | 69.56 |
| | | LEMoE | 34.74 | 33.43 | 32.05 | 55.55 | 50.04 | 41.16 | 67.97 | 61.07 | 58.16 | 48.48 | 44.06 | 55.95 |
| | | LiveEdit | 97.00 | 91.92 | **87.53** | **100.00** | **100.00** | 95.29 | **92.22** | 83.97 | **82.75** | **100.00** | **100.00** | **91.79** |
| | | Ours | **97.38** | **94.46** | 86.85 | **100.00** | **100.00** | **95.74** | 91.97 | **84.25** | 82.68 | **100.00** | **100.00** | 91.68 |

