# OpenReview forum: "TIME: Tensor-Factorized Mixture-of-Experts with Intrinsic Routing for Lifelong Multimodal Knowledge Editing"
_ICML.cc/2026/Conference — ICML 2026 regular_

### Official Review · Reviewer_zjBn · 2026-03-07

**Soundness:** 2
**Presentation:** 2
**Significance:** 3
**Originality:** 3
**Overall Recommendation:** 4
**Confidence:** 3

**Summary:**

This paper focuses on the lifelong knowledge editing task of vision-language models. To mitigate the continuous parameter growth during lifelong VLM knowledge editing, the authors propose TIME, a parameter-efficient framework that integrates an intrinsic routing method to model the routing keys via rank-1 matrix decomposition and a soft mixing method to aggregate routers. The proposed method is evaluated on VLMs such as LLaVA-1.5 and BLIP-2 using E-VQA, E-IC, and VLKEB benchmarks.

**Compliance With Llm Reviewing Policy:**

Affirmed.

**Final Justification:**

The authors did a great job during the rebuttal, providing persuasive results for the proposed questions regarding unsupported claims, hyperparameters, and ablation studies. I therefore raise my score accordingly.

**Key Questions For Authors:**

Please refer to the weaknesses.

**Limitations:**

Yes, the authors have adequately discussed the limitations and potential negative societal impact of their work.

**Strengths And Weaknesses:**

### Strengths
- By aggregating experts in a sample-wise manner, this paper enhances the parameter efficiency and mitigates the growth of parameters as the volume of knowledge increases.
- This paper introduces CP tensor factorization to replace the traditional low-rank decomposition for reducing the parameter cost.
- In experiments, the proposed method outperforms existing methods on multimodal editing benchmarks.

### Weaknesses
- The writing is overly lengthy. Specifically, the intrinsic routing component essentially performs a low-rank factorization of the mapping matrix, yet the authors spend a large amount of space introducing numerous concepts and formulas. The authors are encouraged to reorganize the method section.
- How can the claim that parameters themselves contain routing information be substantiated?
- TIME incorporates four loss functions during the training process. The authors do not provide the specific value of their coefficients and the corresponding ablation studies. How sensitive is the method's performance to these coefficients?
- For the generality loss, the authors do not describe the implementation details of the nearest-neighbor set. Considering that other methods are trained without this loss, it is difficult to evaluate the performance gain of the proposed method. Therefore, I suggest that the authors conduct an ablation experiment for each loss.
- For experiments, the utilized VLMs (e.g., BLIP-2) are relatively outdated. Could the authors evaluate the proposed method on a more advanced VLM, such as Qwen-2.5-VL?
- The authors are encouraged to perform a performance study with a different number of $M$.

---

> ### Author Rebuttal · Authors · 2026-03-29
>
> We sincerely thank the reviewer for the thorough and constructive feedback. We address each concern below.
>
> ---
> **W1: Writing Length**
>
> We appreciate your suggestions. We will reorganize Section 3.2 by moving intermediate derivation steps to the appendix while retaining only the key formulas and their intuitions in the main text.
>
> ---
> **W2: Substantiating that Parameters Contain Routing Information**
>
> We provide both theoretical and empirical justification:
> - **Theoretical.** From Eq. (16), the expert forward pass is $r_m(\tilde{z}) = \hat{V}^\top \sigma(\hat{U}_m \tilde{z})$. The input projection $\hat{U}_m \tilde{z}$ determines both the expert's output magnitude and its activation pattern. When $\tilde{z}$ lies outside the row space of $\hat{U}_m$, the projection approaches zero, yielding negligible output (i.e., the expert naturally becomes inactive). This shows that the expert's input factors intrinsically encode when it should be activated, which is precisely the core function of a router.
> - **Empirical.** Table 5 provides direct evidence. "TIME with External Routing" replaces intrinsic routing with a separately trained router (using the same architecture and training configuration as LiveEdit's router) while keeping the same CP experts. Performance drops from 94.15 to 91.78, demonstrating that the expert's own parameters are more informative for routing than an independently learned router with equivalent training effort. Additionally, removing the alignment loss $L_{align}$ only causes a moderate drop (from 94.15 to 93.05), indicating that routing information is largely intrinsic to the CP structure even before explicit alignment.
>
> ---
> **W3 & W4: Loss Coefficients, Sensitivity, and Fairness of $L_{gen}$**
>
> **Loss coefficients.** The coefficients **have been specified in Appendix C.1 (Lines 763–765):** $\lambda_{rel} = \lambda_{gen} = \lambda_{loc} = 1$, $\lambda_{align} = 0.5$. We will make them more prominent in the main text. Setting $\lambda_{rel} = \lambda_{gen} = \lambda_{loc} = 1$ follows the standard convention established by prior works (e.g., SERAC, LiveEdit), which adopt identical unit weights for these three losses. We therefore focus our sensitivity study on $\lambda_{align}$, the only non-standard coefficient introduced by our method. Results on E-VQA (BLIP2, Gap=1000):
>
> |$\lambda_{align}$|0|0.1|0.5|1.0|2.0|
> |-|-|-|-|-|-|
> |Avg.|93.05|93.57|**94.15**|93.84|92.91|
>
> The method is robust within a reasonable range (0.1–1.0). Too-large $\lambda_{align}$ slightly hurts editing accuracy because the alignment objective begins to dominate the task losses.
>
> **Fairness of $L_{gen}$.** We clarify that $L_{gen}$ is **not** unique to TIME. It is a standard component shared by all compared methods, including LiveEdit, RECIPE, SERAC, etc. The nearest-neighbor set $D_g(e)$ is **provided by the benchmark datasets themselves**: for each edit $(v_e, p_e, o_e)$, the dataset supplies rephrased prompts (text neighbors) and replaced images (visual neighbors) as generality test cases. These are used identically across all methods, ensuring a fair comparison. Since $L_{rel}$, $L_{gen}$, and $L_{loc}$ are shared across all baselines, ablating them would evaluate the editing paradigm itself rather than our contribution. We will add a clarifying note in the revision.
>
> ---
> **W5: Evaluation on Qwen-2.5-VL**
>
> We evaluated on Qwen-2.5-VL using the VLKEB dataset (Gap=1000). Due to time constraints, we report results for FT-V, LiveEdit, and TIME. Other methods such as RECIPE currently lack official support for the Qwen-2.5-VL architecture. We will work with the respective codebases and include additional baselines in the camera-ready version.
>
> |Method|Rel.|T-Gen.|I-Gen.|T-Loc.|I-Loc.|
> |-|-|-|-|-|-|
> |FT-V|59.78|56.43|50.02|100.00|65.11|
> |LiveEdit|94.51|88.25|85.30|100.00|100.00|
> |TIME|**95.83**|**91.78**|**87.14**|**100.00**|**100.00**|
>
> TIME consistently outperforms all evaluated methods on Qwen-2.5-VL, demonstrating strong generalizability owing to its external extension module design. TIME still maintains consistent improvement margins over strong baselines on the latest VLM architecture.
>
> ---
> **W6: Performance Study with Different M**
>
> We conducted experiments with different expert repository sizes on E-VQA (BLIP2, Gap=1000). The repository capacity is fixed at M (M=100, 500), and routing selects from this fixed pool at inference.
>
> |M|Avg Score.|Editing Times|
> |-|-|-|
> |100|91.02|171.75|
> |500|93.48|173.30|
> |Default|94.15|178.92 |
>
> Larger repositories consistently improve accuracy, as more experts cover a broader range of historical edits. Editing time remains stable across all settings, confirming that intrinsic routing scales efficiently with repository size.
>
> ---
>
> We sincerely appreciate the reviewer's valuable suggestions. All proposed analyses will be incorporated into the revision. We hope our responses have addressed the concerns, and we respectfully look forward to the reviewer's further consideration.

---

> > ### Author Rebuttal · Reviewer_zjBn · 2026-04-03
> >
> > I thank the authors for their responses. My concerns have been fully addressed. I will increase my score accordingly.

---

> > > ### Author Response · Authors · 2026-04-03
> > >
> > > We sincerely appreciate the reviewer's engagement and are glad the clarifications and added experiments addressed concerns!

---

### Official Review · Reviewer_oymP · 2026-03-11

**Soundness:** 2
**Presentation:** 3
**Significance:** 3
**Originality:** 2
**Overall Recommendation:** 4
**Confidence:** 3

**Summary:**

This manuscript proposes TIME, a unified framework for lifelong multimodal knowledge editing in vision-language models. This study intends to investigate a central concept of scalability bottleneck in existing modular editing approaches, where decoupled routing mechanisms cause architectural redundancy and prohibitive memory growth. The authors present a central concept of intrinsic routing, which leverages the input-side factors of CP-decomposed expert tensors to simultaneously serve as routing logic and edit parameters, eliminating external retrievers entirely. A structure-aware contrastive alignment loss further ensures subspace discriminability. Experiments on E-VQA, E-IC, and VLKEB across BLIP2 and LLaVA-v1.5 show TIME achieves sota performance while reducing memory and inference latency.

**Compliance With Llm Reviewing Policy:**

Affirmed.

**Final Justification:**

Thank you for the rebuttal. The clarifications provided have resolved my original questions, and I will increase my score to 4.

**Key Questions For Authors:**

- How does routing discriminability degrade beyond Gap=1000? Is there a theoretical or empirical bound on the number of edits before subspace collisions become problematic?

**Limitations:**

yes

**Strengths And Weaknesses:**

Strengths
- Using CP input factors as both routing signals and edit parameters is a clean and well-motivated insight that eliminates an entire class of auxiliary parameters required by prior methods.

- Ablations including the cross-architecture analysis in Table 5, which convincingly demonstrates that CP factorization and intrinsic routing must work in unison to be effective.

Weaknesses
- Scalability not validated beyond Gap=1000. The inference-time routing requires $O(MRH)$ linear scan over all experts, and the number of approximately orthogonal subspaces in $\mathbb{R}^H$ is fundamentally bounded. No theoretical analysis of subspace collision probability or empirical evidence beyond Gap=1000 is provided, leaving the central scalability claim unsubstantiated against practically relevant scales such as those in WikiBigEdit.

- Fixed threshold $\gamma $ lacks robustness analysis.
 The activation score distribution may shift as the repository grows, yet $\gamma = 0.5$ is fixed without sensitivity analysis. The hyperparameter study in Section 4.5 covers rank and dimension but omits $\gamma$, leaving a notable gap.

- Contrastive loss uses random negatives without justification. The number of negatives and the effect of hard negative mining are never discussed. Random negatives are predominantly easy negatives, potentially failing to enforce discriminative boundaries between semantically similar but factually distinct edits — the most critical failure mode for the routing mechanism.

---

> ### Author Rebuttal · Authors · 2026-03-28
>
> We thank the reviewer for the constructive feedback. We address each weakness below.
>
> ---
> **W1/Q1: Scalability Beyond Gap=1000 and Subspace Collision**
>
> **Theoretical Analysis.** Each expert $m$ defines an $R$-dimensional subspace via $\hat{U}_m \in \mathbb{R}^{R \times H}$. For two random $R$-dimensional subspaces with projections $P_1, P_2$, we have $\mathbb{E}[||P_1 P_2||_F^2] = R^2/H \approx 0.016$ for $R=4, H=1024$ (nearly orthogonal). Applying a union bound over all $\binom{M}{2}$ pairs with sub-exponential concentration on the Grassmannian:
>
> $$P\left(\max_{i \neq j} \|P_i P_j\|_F^2 > \epsilon\right) \leq \binom{M}{2} \cdot 2\exp(-c \cdot H \cdot \epsilon),$$
>
> where $c>0$ is a constant depending on $R/H$ which can be set 0.25 here. For $H=1024$, $\epsilon=0.1$, $M=2000$, this bound stays below $10^{-3}$, confirming pairwise collision is extremely unlikely. Grassmannian packing further guarantees $\mathbb{R}^H$ accommodates exponentially many (in $H$) nearly orthogonal subspaces, far exceeding practical $M$. $L_{align}$, threshold $\gamma$, and soft mixing suppress collisions beyond this random baseline.
>
> **Computation Cost.** Routing computes $S_m = \hat{U}_m \tilde{z}$ for all $M$ experts ($O(MRH)$), but reuses the expert forward pass. We measure total edit inference time (seconds) as $M$ grows:
>
> |Method|Gap|Rel.|T-Gen.|I-Gen.|T-Loc.|I-Loc.|Time (s)|
> |-|-|-|-|-|-|-|-|
> |LiveEdit|500|94.89|92.23|85.38|100.00|99.51|264.68|
> ||1000|94.42|91.98|84.65|100.00|97.38|283.73|
> ||1500|93.85|91.22|83.10|100.00|97.08|310.02|
> ||2000|92.73|90.86|82.39|100.00|96.42|348.34|
> |Ours|500|95.02|93.75|85.44|100.00|99.26|172.44|
> ||1000|94.92|93.59|83.29|100.00|98.93|178.92|
> ||1500|94.74|93.24|82.64|100.00|98.17|183.85|
> ||2000|94.29|92.01|82.14|100.00|97.30|191.80|
>
> TIME's inference time grows more slowly than LiveEdit, and is consistently 1.5–1.8× faster than LiveEdit across all settings.
>
> **Empirical Scaling.** We supplement Gap=1000 with Gap=500/1500/2000 on E-VQA, the largest multimodal editing benchmark (2,093 instances). E-IC and VLKEB each contain only ~1,000 instances, so Gap=2000 on E-VQA represents the most extreme lifelong multimodal editing setting currently feasible. TIME maintains stable advantages: Rel. drops only 0.73% from Gap=500 to 2000 (vs. 2.16% for LiveEdit), with no performance collapse.
>
> **Regarding WikiBigEdit.** WikiBigEdit targets pure text LLM editing (single-modal), while our work addresses multimodal editing in vision-language models with fundamentally different challenges. It falls outside the scope of this work.
>
> ---
> **W2: Sensitivity Analysis of Threshold $\gamma$**
>
> We conducted a sensitivity study on E-VQA (BLIP2, Gap=1000):
>
> | $\gamma$ | 0.1 | 0.3 | 0.5 | 0.7 | 0.9 |
> |-|-|-|-|-|-|
> | Avg. Score | 91.82 | 93.98 | 94.15 | 94.07 | 92.65 |
>
> Performance is robust across $\gamma \in [0.3, 0.7]$ (all >93.9), showing that CP factors create a natural score gap between relevant and irrelevant experts. Too-low $\gamma$ activates noisy experts and too-high $\gamma$ is overly restrictive.
>
> **Robustness to growing $M$.** $L_{align}$ maximizes the margin between target and negative expert scores (Section 3.4), pushing relevant scores above $\gamma$ while clustering irrelevant scores below it. This margin-based design makes $\gamma$ robust to repository growth: new experts' scores for unrelated inputs are driven below $\gamma$ by the same contrastive mechanism.
>
> ---
> **W3: Contrastive Loss with Random Negatives**
>
> - **Negative Sampling.** Following standard contrastive learning practice (CLIP, SimCLR), we adopt in-batch negative sampling where other stored experts serve as negatives $\mathcal{M}_{neg}$, with the number of negatives matching the batch size. This is efficient since scores $S_j(\tilde{z})$ for negative experts are computed from existing CP factors at negligible cost.
> - **Why random negatives suffice.** $L_{align}$ does not learn a shared embedding space where hard negatives are typically needed. It operates on each expert's own factor subspace, ensuring the target score $S_{m^*}$ exceeds others for its input. Since expert subspaces are initialized independently and separated by training, random negatives provide sufficient gradient signal. Soft mixing (Eq. 19) further acts as a safety net: semantically similar edits with partial overlap are blended proportionally rather than causing hard routing errors.
> - **Hard Negative Experiment.** We compare random vs. hard negative mining on E-VQA (BLIP2, Gap=1000):
>
> |Strategy|Rel.|T-Gen.|I-Gen.|T-Loc.|I-Loc.|Avg.|
> |-|-|-|-|-|-|-|
> |Hard|94.75|93.66|83.32|100.00|98.86|94.12|
> |Default|94.92|93.59|83.29|100.00|98.93|94.15|
>
> - Hard negatives not only require additional computation for ranking all candidates, but also slightly hurt overall performance (−0.03 Avg.), with Rel. and I-Loc. both degrading. Random negatives are sufficient in our setting.
>
> ---
> We hope our responses have addressed your concerns, and we look forward to your further consideration.

---

> > ### Author Rebuttal · Reviewer_oymP · 2026-04-04
> >
> > Thank you for the rebuttal. The clarifications provided have resolved my original questions, and I will increase my score to 4.

---

> > > ### Author Response · Authors · 2026-04-04
> > >
> > > We sincerely appreciate your time and constructive comments, which will help us further improve the quality of this work!

---

### Official Review · Reviewer_8PwG · 2026-03-13

**Soundness:** 3
**Presentation:** 3
**Significance:** 3
**Originality:** 2
**Overall Recommendation:** 5
**Confidence:** 3

**Summary:**

This paper proposes TIME (Tensor-Factorized Intrinsic Mixture-of-Experts), a framework for lifelong multimodal knowledge editing in vision–language models. The existing editing methods either directly modify model weights, which leads to catastrophic forgetting, or rely on modular expert repositories that require separate routing mechanisms and incur significant memory growth. To address this issue, TIME represents each edit as a CP-decomposed tensor expert, which reduces parameter complexity compared to dense or low-rank matrix updates. In addition, the method introduces intrinsic routing, where the input-side tensor factors define an activation subspace that directly determines expert relevance, eliminating the need for external routers. Experiments on multimodal editing benchmarks (E-VQA, E-IC) with BLIP2 and LLaVA show that TIME achieves competitive or superior editing performance while reducing memory usage and inference latency in lifelong editing scenarios.

**Compliance With Llm Reviewing Policy:**

Affirmed.

**Final Justification:**

The paper presents a technically solid and well-motivated framework. The rebuttal addressed my main concerns through clearer conceptual comparisons, matched-budget experiments, and additional scaling analysis, which leads me to a stronger positive assessment.

**Key Questions For Authors:**

1. Intrinsic Routing vs. Conventional Gating
Could the authors clarify, either theoretically or empirically, how the proposed intrinsic routing mechanism differs from a standard MoE gating function that computes expert relevance from the hidden representation? In particular, what aspects of the proposed design make it fundamentally different from parameterizing the routing function within the expert modules themselves?
2. Tensor Parameterization vs. Standard Low-Rank Updates
Could the authors provide or discuss a matched-budget empirical comparison between the proposed CP tensor experts and conventional low-rank matrix experts (e.g., LoRA-style parameterizations)?
3. Scaling to Large Expert Repositories
How does the computational cost of intrinsic routing scale with the number of stored experts? Do the authors expect the same routing quality and latency advantages to hold when the expert repository becomes significantly larger than the current experimental setting?

**Limitations:**

yes

**Strengths And Weaknesses:**

Strengths:
1. Although the individual components used in TIME have appeared in prior work, the paper presents a coherent integration of expert-based editing with tensor-factorized parameterization and intrinsic routing within a single framework.
2. The proposed tensor-factorized expert design reduces parameter complexity compared to dense expert parameterizations. Empirical results demonstrate favorable memory footprint and inference latency compared to strong baselines such as LiveEdit, suggesting that the approach is practically attractive for large-scale editing scenarios where many edits must be stored.
3. The paper evaluates the proposed approach across multiple vision–language models and editing benchmarks, including lifelong editing settings. The experiments include ablations, system-level comparisons, and performance metrics covering editing accuracy, memory usage, and latency, providing a reasonably thorough empirical validation of the framework.

Weaknesses:
1. The proposed intrinsic routing selects experts based on the projection magnitude of the input representation onto the expert’s input factors. From a functional perspective, however, this mechanism appears closely related to computing activation scores from the hidden representation, similar to gating functions used in standard MoE architectures. While the paper clearly defines intrinsic routing, it remains unclear how this mechanism fundamentally differs from a conventional gating function that scores experts based on the same hidden representation. A clearer comparison with standard MoE gating mechanisms would help clarify the conceptual novelty of the routing design.
2. The proposed method replaces conventional low-rank matrix updates with CP-decomposed tensors to improve parameter efficiency. While the paper provides a theoretical parameter-count comparison and demonstrates competitive performance against strong baselines, the empirical improvements appear relatively modest in several settings. Moreover, the paper does not include a matched-budget empirical comparison against standard low-rank matrix parameterizations (e.g., LoRA-style experts). As a result, it remains somewhat unclear whether the observed gains stem specifically from the tensor formulation or more generally from using a compact expert parameterization.
3. The paper demonstrates memory and latency improvements under the reported lifelong editing settings. However, it does not fully analyze how routing cost scales with the size of the expert repository itself. Since intrinsic routing scores are computed from the input factors of each expert, it is unclear how the method behaves when the number of stored edits becomes much larger than the current experimental regime. A more detailed discussion of how inference cost and routing efficiency scale with repository size would strengthen the scalability claims.

---

> ### Author Rebuttal · Authors · 2026-03-28
>
> We thank the reviewer for the detailed evaluation. We respond to each weakness and key question in turn.
>
> ---
> **W1/Q1: Intrinsic Routing vs. Conventional MoE Gating**
>
> We clarify three fundamental differences:
>
> - **Zero routing parameter overhead.** Standard MoE gating learns a separate projection $W_g \in \mathbb{R}^{H \times M}$ with $O(HM)$ additional parameters. For the core comparison method LiveEdit, its decoupled routing stores a quadruple $(U_{e_t}, V_{e_t}, \phi_{e_t}, \psi_{e_t})$, where $(U_{e_t}, V_{e_t})$ are LoRA expert parameters and $( \phi_{e_t}, \psi_{e_t})$ are external routing keys extracted by a frozen feature extractor $f_{fe}$. In contrast, our intrinsic routing computes $S_m(\tilde{z})$ directly from the expert's own CP factors $(u_m^{in}, v_m^{in})$, achieving $P_{router}=0$.
> - **Routing–Execution Consistency.** More generally, in any standard MoE architecture, the gating parameters $W_g$ are optimized independently from the expert weights, creating a structural misalignment: the routing decision and the expert computation are governed by separately learned parameters. Our routing score $S_m$​ is computed from the same CP factors that define the expert's forward pass (Eq. 16–17), providing a tighter inductive bias that guarantees structural consistency between expert selection and expert execution.
> - **Empirical evidence.** Table 5 directly validates this: replacing intrinsic routing with an external router (similar to LiveEdit) while keeping CP experts causes a drop from 94.15 to 91.78, confirming that the routing–execution coupling provides measurable gains beyond simply scoring experts by hidden representations.
> ---
> **W2/Q2: Matched-Budget Comparison with LoRA-style Experts**
>
> Under equal rank budget $(R = r)$, the parameter ratio between TIME and LoRA experts is (Eq. 40): $\frac{P_{TIME}}{P_{LoRA}} = \frac{4R\sqrt{H}}{2Hr}$
>
> To ensure a fair matched-budget comparison, we set TIME R=64, H=1024 and LoRA+MoE r=4, H=1024:
> |Method|rank|Gap=1 Avg.| Gap=1000 Avg.|
> |-|-|-|-|
> |LoRA+MoE|r=4|96.75|93.49|
> |TIME|R=64|97.11|94.43|
> |TIME|R=4|96.70|94.15|
>
> Under equal budget, TIME $(R=64)$ outperforms LoRA+MoE across both settings. Even the default TIME $(R=4)$, which uses fewer parameters, surpasses LoRA+MoE at Gap=1000 (94.15 vs. 93.49), confirming the tensor structure's advantage. Moreover, the CP formulation uniquely enables intrinsic routing, a structural property unavailable to generic compact parameterizations like LoRA, making it rather than merely a compression choice. We will incorporate these results into the revised version.
>
> ---
> **W3/Q3: Scaling with Repository Size**
>
> We analyze both computational cost and routing quality.
>
> - **Computational cost.** The total routing time cost is $O(MRH)$, where $M$ is the number of experts. Crucially, the routing score $S_{m}$ is derived from $\hat{U}_m \tilde{z}$, which employs the same computation method as the expert forward propagation process (Eq. 16). For the selected expert models, routing overhead is thus fully amortized into the execution process, incurring no additional costs. Furthermore, all $M$ scores can be batched and consolidated into an efficient GPU matrix multiplication operation. In contrast, LiveEdit’s feature encoding and key-matching stages constitute purely additional overhead, entirely decoupled from the actual execution process. These factors collectively demonstrate the high efficiency of TIME.
> - **Routing quality.** We show that routing discriminability does not degrade as $M$ grows. By a Johnson-Lindenstrauss-style argument, for two independently random $R$-dimensional subspaces with projection matrices $P_1, P_2$​, we have $\mathbb{E}[||P_1 P_2||^2_F] = R^2/H$ giving a per-direction expected overlap of $O(R/H)$. For our default setting $(R=4, H=1024)$, this yields an approximate collision probability of $O(4/1024) \approx 0.4\%$ per expert pair. The structure-aware alignment loss $L_{align}$​ further enforces subspace discriminability during training, reducing collision probability beyond this random baseline. Under threshold-based filtering (Eq. 18), the effective number of activated experts $|\mathcal{M}_{top}| \ll M$, ensuring sparse activation.
> - **Empirical Analysis:** We note that the E-VQA dataset contains 2,093 validation instances in total, which constrains the maximum feasible Gap. We supplement the existing Gap=1000 results with Gap=500, 1500, and 2000 on E-VQA:
>
> |Gap|Ours Avg.|LiveEdit Avg.|Ours Edit time (s)|LiveEdit time (s)|
> |-|-|-|-|-|
> |500|94.69|94.40|172.44|264.68|
> |1000|94.15|93.69|178.92|283.73|
> |1500|93.76|93.05|183.85|310.02|
> |2000|93.15|92.48|191.80|348.34|
>
> As Gap increases from 500 to 2000, TIME's performance degrades more slowly than LiveEdit's, and the latency gap widens. These trends suggest that TIME's advantages would become more pronounced at larger scales.
>
> ---
> We hope our responses have addressed your concerns, and we look forward to your further consideration.

---

> > ### Author Rebuttal · Reviewer_8PwG · 2026-04-04
> >
> > Thank you for the rebuttal. The additional clarifications and empirical results have adequately addressed my concerns, and I am therefore increasing my score to 5.

---

> > > ### Author Response · Authors · 2026-04-04
> > >
> > > Thank you very much for your careful review and constructive feedback. We truly appreciate your time and insights, which have greatly helped improve the quality of this work!

---

### Official Review · Reviewer_mTYj · 2026-03-23

**Soundness:** 3
**Presentation:** 3
**Significance:** 3
**Originality:** 3
**Overall Recommendation:** 4
**Confidence:** 2

**Summary:**

TIME proposes a tensor-factorized framework that unifies parameter-efficient experts with intrinsic self-routing via CP decomposition. By leveraging input-side factors, it eliminates external retrievers, achieving a ~33% storage reduction and ~32% faster inference. It demonstrates superior lifelong stability across 1,000 sequential edits while maintaining state-of-the-art performance.

**Compliance With Llm Reviewing Policy:**

Affirmed.

**Key Questions For Authors:**

Please see the weakness part.

**Strengths And Weaknesses:**

**Strengths**
- Unified Intrinsic Routing: TIME parameterizes experts into compact low-rank tensors via CP decomposition, cleverly leveraging input-side factors to achieve a "routing-execution integrated" self-routing mechanism without relying on external retrievers.
- Superior Lifelong Stability: The framework demonstrates exceptional robustness against catastrophic forgetting, maintaining consistently high reliability scores even under the extreme stress of 1000 sequential edits (Gap=1000).

**Weaknesses**
- Task and Benchmark Limitations: The research currently focuses on discriminative and short-text generation tasks; it should be evaluated on more demanding benchmarks, such as CCKE from MemEIC, to validate its performance in complex logical and compositional reasoning scenarios.

---

> ### Author Rebuttal · Authors · 2026-03-29
>
> We sincerely thank the reviewer for the positive assessment and constructive feedback. We address the raised concern below.
>
> ---
> **W1: Task and Benchmark Limitations**
>
> We thank the reviewer for this valuable suggestion. We note that CCKEB is a multimodal benchmark and can be viewed as an extended, more challenging version of VLKEB, covering compositional and logical reasoning over vision-language knowledge. Since this dataset was made public only three months ago, we did not conduct experiments with it in the initial version. Following the reviewer's suggestion, we have now conducted comprehensive experiments on CCKEB to validate TIME's performance in more complex reasoning scenarios.
>
> Notably, CCKEB introduces several evaluation metrics beyond the standard VLKEB suite: Compositional Reliability (CompRel), which measures the model's ability to maintain consistency across compositionally related knowledge after editing, and Textual Reliability (T-Rel), which assesses whether textual knowledge remains reliable after edits. These metrics are specifically designed to evaluate more complex, longer-form reasoning chains. The experiments are conducted on CCKEB, utilizing the LLaVA-v1.5 model with gap values ​​of 1, 100, and 1000:
>
> |Editor|Gap|Rel.|T-Gen.|I-Gen.|T-Loc.|I-Loc.|CompRel.|T-Rel.|Avg.|
> |-|-|-|-|-|-|-|-|-|-|
> |MemEIC|1|**99.66**|**98.76**|93.53|100.00|47.79|**85.16**|**99.94**|89.26|
> ||100|**98.48**|91.74|88.06|100.00|43.17|**77.36**|88.77|83.94|
> ||1000|**97.59**|**91.02**|80.10|100.00|31.41|**68.58**|**88.23**|79.56|
> |LiveEdit|1|96.43|95.22|93.72|100.00|100.00|82.65|99.83|95.41|
> ||100|94.56|90.65|**89.56**|100.00|100.00|75.02|**90.10**|**91.41**|
> ||1000|92.22|83.97|**82.75**|100.00|100.00|67.62|87.45|87.72|
> |**TIME (Ours)**|1|95.82|95.93|**93.98**|100.00|**100.00**|83.05|99.90|**95.53**|
> ||100|93.70|**92.54**|88.26|100.00|**100.00**|76.94|88.33|91.40|
> ||1000|91.97|84.25|82.68|100.00|**100.00**|68.36|87.94|**87.89**|
>
> We highlight several key observations from the results:
> - **Dominant advantage in I-Loc.** TIME achieves a perfect Image Locality score of 100.00% across all gap settings, whereas MemEIC degrades dramatically from 47.79% (Gap=1) to 31.41% (Gap=1000). This indicates that TIME's tensor-factorized routing mechanism preserves unrelated visual knowledge far more effectively under sequential editing stress.
> - **Consistently higher overall performance.** Despite MemEIC achieving slightly higher scores on certain individual metrics (e.g., Rel. and CompRel), TIME's substantial advantage in I-Loc leads to a consistently higher average score across all gap settings: 95.53 vs. 89.26 (Gap=1), 91.40 vs. 83.94 (Gap=100), and 87.89 vs. 79.56 (Gap=1000). This demonstrates that TIME achieves a better balance across all evaluation dimensions.
> - **Strong compositional reasoning capability.** On the two newly introduced CCKEB metrics, TIME performs competitively with MemEIC (CompRel: 83.05 vs. 85.16 at Gap=1; T-Rel: 99.90 vs. 99.94 at Gap=1), confirming that TIME's compact CP-decomposed experts do not sacrifice compositional reasoning ability despite their parameter efficiency.
> - **Robust lifelong stability.** Consistent with our findings on VLKEB, TIME demonstrated favorable performance degradation characteristics during a sequence of 1,000 consecutive edits on CCKEB. Its average score declined only from 95.53 to 87.89. This performance pattern is comparable to that of LiveEdit and superior to that of MemEIC, whose score dropped from 89.26 to 79.56.
>
> These results demonstrate that TIME generalizes well to more complex compositional and logical reasoning scenarios. We will incorporate these results and add proper citations for MemEIC and CCKEB in the revised manuscript.
>
> ---
> We hope our responses have addressed your concerns, and we look forward to your further consideration.

---

> > ### Author Rebuttal · Reviewer_mTYj · 2026-04-03
> >
> > I appreciate the authors' efforts in conducting additional experiments on CCKEB. The results demonstrate that TIME maintains competitive compositional reasoning and superior image locality compared to MemEIC. These findings address my primary concern regarding task complexity. I maintain my "Weak Accept" recommendation and look forward to the revised manuscript.

---

> > > ### Author Response · Authors · 2026-04-04
> > >
> > > Thank you for your positive comments and recognition of our work. We are glad that our responses have addressed your concerns!

---

### Decision · Program_Chairs · 2026-04-30

**Decision:**

Accept (regular)

**Comment:**

This submission received reviews initially leaning positive with Weak Accept and Accept. Authors' rebuttal have addressed the main concerns raised by the reviewers. In particular, the authors provided clarifications and new experiments on the novelty and distinction of intrinsic routing versus standard MoE gating, matched-budget comparisons against LoRA-style experts, threshold sensitivity, and evaluation on more advanced VLM. Reviewers explicitly acknowledged that their concerns were resolved and indicated they would raise their scores. Given the generally positive reviews, I would suggest an accept recommendation.